
# The characteristics of the 2022 Tonga volcanic tsunami in the Pacific Ocean

Gui Hu[1], Linlin Li[1,2], Zhiyuan Ren[3], Kan Zhang[1]

1. Guangdong Provincial Key Laboratory of Geodynamics and Geohazards, School of Earth Sciences and Engineering, Sun Yat-sen University, Guangzhou, China

2. Southern Marine Science and Engineering Guangdong Laboratory (Zhuhai), Zhuhai, China

3. Department of Civil and Environmental Engineering, National University of Singapore, Singapore.

*Correspondence to:* Linlin Li (lilinlin3@mail.sysu.edu.cn)

**Abstract.** On 15th January 2022, an exceptional eruption of Hunga Tonga–Hunga Ha'apai volcano generated atmospheric and tsunami waves that were widely observed at oceans globally, gaining a remarkable attention to scientists in related fields. The tsunamigenic mechanism of this rare event remains an enigmatic due to its complexity and lacking of direct underwater observations. Here, to explore the tsunamigenic mechanisms of this volcanic tsunami event and its hydrodynamic processes in the Pacific Ocean, we conduct tsunami waveform and spectral analyses of the waveform recordings at 116 coastal gauges and 38 deep-ocean buoys across the Pacific Ocean. Combined with the constraints of some representative barometers, we obtain the plausible tsunamigenic origins during the volcano activity. We identify four distinct tsunami wave components generated by air-sea coupling and seafloor crustal deformation. Those tsunami components are differentiated by their different propagating speeds or period bands. The first-arriving tsunami component with ~80–100 min period was from shock waves spreading at a velocity of ~1000 m/s in the vicinity of the eruption. The second component with extraordinary tsunami amplitude in deep sea was from Lamb waves. The Lamb wave with ~30–40 min period radically propagated outward from the eruption site with spatially decreasing propagation velocities from ~340 m/s to ~315m/s. The third component with ~10–30 min period was probably from some atmospheric gravity wave modes propagating faster than 200 m/s but slower than Lamb waves. The last component with ~3–5 min period originated from partial caldera collapse with dimension of ~0.8–1.8 km. Surprisingly, the 2022 Tonga volcanic tsunami produced long oscillation in the Pacific Ocean which is comparable with those of the 2011 Tohoku tsunami. We point out that the long oscillation is not only associated with the resonance effect with the atmospheric acoustic-gravity waves, but more importantly the interactions with local bathymetry. This rare event also calls for more attention to the tsunami hazards


produced by atypical tsunamigenic source, e.g., volcanic eruption.

## 1. Introduction

On 15 January 2022 at 04:14:45 (UTC), a submarine volcano erupted violently at the uninhabited Hunga
Tonga-Hunga Ha'apai (HTHH) island at 20.546°S 175.390°W (USGS, 2022). The volcano is located ~67
km north of Nuku'alofa, the capital of Tonga (NASA, 2022) (Figure 1). The blasts launched plumes of
ash, steam, and gas ~58 km high into stratosphere (Yuen et al., 2022) which not only blanketed nearby
islands in ash (Duncombe, 2022; NASA, 2022), but caused various atmospheric acoustic-gravity wave
modes (AGWs) of various scales, e.g., Lamb waves from atmospheric surface pressure disturbance
associated with the eruption (Liu and Higuera, 2022; Adam, 2022; Kubota et al., 2022; Matoza et al.,
2022). Tsunami with conspicuous sea level changes were detected by coastal tide gauges and Deep-ocean
Assessment and Reporting of Tsunamis (DART) buoy stations in the Pacific (Figure 1), the Atlantic, and
Indian Oceans as well as the Caribbean and Mediterranean seas (Carvajal et al., 2022; Kubota et al., 2022;
Ramírez-Herrera et al., 2022), while the large waves were mainly concentrated in the Pacific Ocean, like
coastlines of New Zealand, Japan, California, and Chile (Carvajal et al., 2022). The event caused at least
3 fatalities in Tonga. Two people drowned in northern Peru when ~2 m destructive tsunami waves
inundated an island in the Lambayeque region, Chile (Edmonds, 2022).
Satellite images revealed that the elevation of HTHH island has gone through dramatic change before
and after the mid-January 2022 eruption. Previously, after the 2015 eruption, the two existing Hunga
Tonga and Hunga Ha'apai Islands were linked together. The volcanic island rose 1.8 km from the seafloor
where it stretched ~20 km across and topped a underwater caldera ~5 km in diameter (Garvin et al., 2018;
NASA, 2022). After the violent explosion on 15 January 2022, the newly formed island during 2015 was
completely gone, with only small tips left in far southwestern and northeastern HTHH island (NASA,
2022). HTHH volcano lies along the northern part of Tonga–Kermadec arc, where the Pacific Plate
subducts under the Indo-Australian Plate (Billen et al., 2003). The convergence rate (15~24 cm/year)
between the Tonga-Kermadec subduction system and the Pacific plate is among the fastest recorded plate
velocity on Earth, forming the second deepest trench around the globe (Satake, 2010; Bevis et al., 1995).
The fast convergence rate contributes to the frequent earthquakes, tsunamis and volcanic eruptions in
this region historically (Bevis et al., 1995). The 2022 HTHH volcano is part of a submarine-volcano
chain that extends all the way from New Zealand to Fiji (Plank et al., 2020). HTHH volcano had many
notable eruptions before 2022 since its first historically recorded eruption in 1912, i.e., in 1937, 1988,
2009, 2014-2015 (Global Volcanism Program, https://volcano.si.edu).

**Figure 1. The spatial distribution of the eruption site (red star), DART stations (squares), tide gauges (triangles) and the calculated tsunami arrival times. White contours indicate the modelled arrival times of conventional tsunami. Red contours indicate the estimated arrival times of Lamb waves (see how we derive these contours in section 3.1).**

The 2022 HTHH eruption is the first volcanic event which generates worldwide tsunami signatures since
the 1883 Krakatau event (Matoza et al., 2022; Self and Rampino, 1981; Nomanbhoy and Satake, 1995).
The tsunamigenic mechanism of this rare volcanic eruption-induced tsunami is still poorly understood
due to its complex nature and the deficiencies of near-field seafloor surveys. Various tsunami generation
mechanisms have been proposed so far based on the observations of ground-based and spaceborne
geophysical instrumentations (Kubota et al., 2022; Matoza et al., 2022; Carvajal et al., 2022). The most-
mentioned mechanism is the fast-traveling atmospheric Lamb wave generated by the atmospheric
pressure rise of ~2 hPa during the eruption. The Lamp wave circled the Earth for several times with
travelling speed close to that of the sound wave in the lower atmosphere, leading to globally observed
sea level fluctuations (Adam, 2022; Duncombe, 2022; Kubota et al., 2022; Matoza et al., 2022) (Figure
1). The second mechanism is suggested to be a variety of other acoustic-gravity wave modes (Adam,
2022; Matoza et al., 2022; Themens et al., 2022; Zhang et al., 2022). The third mechanism may be related



to the seafloor crustal deformation induced by one or more volcanic activities in the vicinity of the
eruption site (e.g., pyroclastic flows, partial collapse of the caldera) (Carvajal et al., 2022) , which are
more responsible for the near-field tsunamis with theoretical tsunami speeds.
To investigate the possible tsunamigenic mechanisms and detailed hydrodynamic behaviors of this rare
volcanic tsunami event, in this study, we collect, process and analyze the sea level measurements from
116 tide gauge and 38 DART buoys in the Pacific Ocean (shown in Figures 1 and 2). We first do statistical
analysis of the tsunami waveforms to estimate the propagating speed of the Lamb wave and to understand
the tsunami wave characteristics in the Pacific Ocean through demonstrating the tsunami wave properties,
i.e., arrival times, wave heights and durations. We then conduct wavelet analysis for representative DART
buoys and tide gauges respectively to explore tsunamigenic mechanisms of the event and to better
understand its hydrodynamic processes in the Pacific Ocean. Aided by wavelet analysis of corresponding
barometers near the selected DART buoys and comparison with tsunami records of the 2011 Tohoku
tsunami, we are able to piece together all the analysis and demonstrate that the 2022 HTHH tsunami was
generated by air-sea coupling with a wide range of atmospheric waves with different propagating
velocities and period bands, and seafloor crustal deformation associated with the volcanic eruption. We
demonstrate as well that the tsunami was amplified at the far-field Pacific coastlines where the local
bathymetric effects play a dominant role in tsunami scale.
**2.    Data and Methods**
**2.1  Data**
We collected high-quality sea level records across the Pacific Ocean at 38 DART buoys (in which 31
stations from https://nctr.pmel.noaa.gov/Dart/, 7 stations from https://tilde.geonet.org.nz/dashboard/) and
116 tide gages from IOC (The Intergovernmental Oceanographic Commission, http://www.ioc-
sealevelmonitoring.org) (Figure 1). The epicentral distances of tide gauges and DART buoys range
between 74–10790 km and 375–10414 km, respectively. The sampling rates of DART buoys are
changing over time. Passing of tsunami event generally can trigger the DART system to enter its high
frequency sampling mode (15 seconds or 1 min) from normal frequency mode (15 min)
(www.ndbc.noaa.gov/dart). In contrast, sampling rates of normal tide gauges at coasts are uniform with
sampling interval of 1 min. The sampling interval of both DART and tide gauges is preprocessed to 15



seconds. Firstly, we eliminate abnormal spikes and fill gaps by linear interpolation. Secondly, we applied
a fourth-order Butterworth-Highpass filter with a cut-off frequency of 3.5 e-5 Hz (~ 8 hours) to remove
the tidal components (Figure 2) (Heidarzadeh and Satake, 2013). After the two steps, quality control step
is conducted to select high-quality data, in which we delete waveforms with spoiled data or massive data
loss due to equipment failure, or with the maximum tsunami heights less than 0.2 m, then the selected
data will be ready for further statistics and spectral analysis. We also collect and analyze the atmospheric
pressure disturbance data recorded by some representative barometers. The sampling rates of the
barometers is generally uniform with a sampling rate of 1 min except for some stations in New Zealand
with interval of 10 min. Considering the sample rate, we employ a fourth-order Butterworth-Bandpass
filter with period ranging between 2–150 min for wavelet analysis of the barometers with 1 min sample
rate, while we apply the fourth-order Butterworth-Bandpass filter with range of 30–150 min to long-
period waveform display based on two reasons. (1) The barometer data we use for the analysis include
some in New Zealand with 10 min sample rate; (2) Filtering out the short-period waves helps highlight
long-period tsunami wave components.
The tsunami waveforms recorded by DART buoys which are installed offshore in the deep water are
expected to contain certain characteristics of the tsunami source (Wang et al., 2020, 2021). The
waveforms recorded by tide gauge distributed along coastlines are significantly influenced by local
bathymetry/topography which are used for investigating bathymetric effect on tsunami behaviors
(Rabinovich et al., 2017, 2006; Rabinovich, 2009). Therefore, we use the DART data for source-related
analysis and choose some tide gauge data to investigate the tsunami behaviors at the Pacific coastlines.



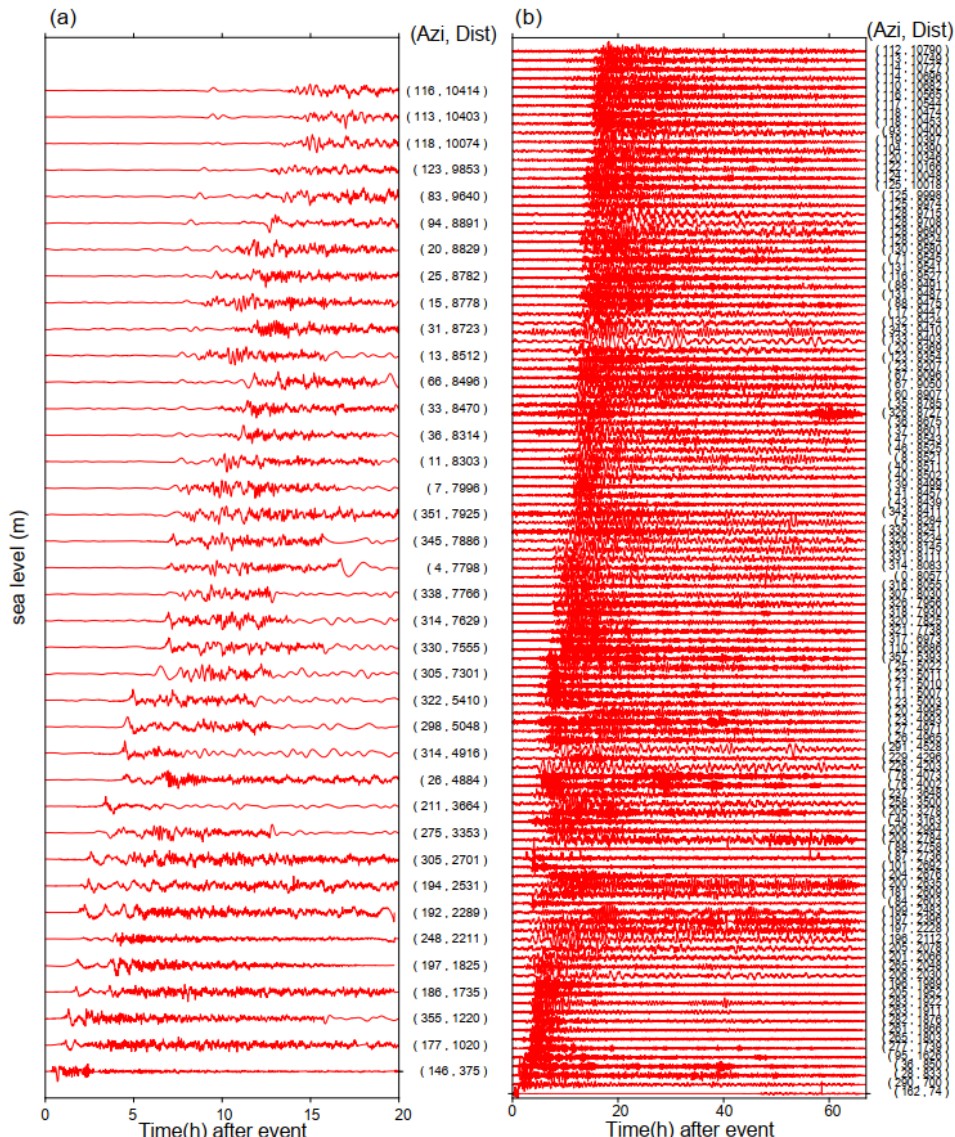

**Figure 2. Detided tsunami waveforms at (a) DART buoys and (b) tide gauges. Waveforms in both subplots are shown in ascending distance.**

## 2.2 Tsunami Modelling

We use a numerical tsunami modelling package JAGURS (Baba et al. 2015) to simulate the tsunami propagation of the 2022 HTHH event and obtain the theoretical tsunami arrival time based on the shallow water wave speed (white contours in Figure 1). The code solves linear Boussinesq-type equations in a spherical coordinate system using a finite difference approximation with the leapfrog method. We specify



a unit Gaussian-shaped vertical sea surface displacement at the volcanic base as the source of
conventional tsunami. For a unite source $i$ with center at longitude $\varphi_i$ and latitude $\theta_i$, the
displacement distribution $Zi(\varphi, \theta)$ can be expressed as:
$$Zi(\varphi, \theta) = exp[-\frac{(\varphi-\varphi_i)^2+(\theta-\theta_i)^2}{2\sigma}]$$ (1)
Where we set characteristic length σ as 5 km (NASA, 2022). The bathymetric data is resampled from the
GEBCO 2019 with 15 arc-sec resolution (The General Bathymetric Chart of the Oceans, downloaded
from https://www.gebco.net).
**2.3  Spectral Analysis of Tsunami Waves**
To investigate the temporal changes of the dominant wave periods, we conduct continuous wavelet
transformation (frequency-time) analyses for some representative DART buoys, tide gauges and
barometers, in which wavelet Morlet mother function is implemented (Kristeková et al., 2006). The first
32-hour time series of DART buoys and barometers after the eruption (at 04:14:45 on 15 January 2022)
are used for source-related wavelet analysis. The first 48-hour time series of tide gauges after the eruption
are employed for hydrodynamics-related wavelet analysis at coastlines. We adopt the Averaged-Root-
Mean-Square (ARMS) method as a measure of absolute average tsunami amplitude with a moving time
window of 20 min to calculate the tsunami duration (Heidarzadeh and Satake, 2014). We define the time
durations as the time period where ARMS levels of tsunami waves are above those prior to the tsunami
arrivals.
**3.   Results**
**3.1  The decreasing propagation velocities of the Lamb Wave**
Although many types of atmospheric waves were generated by the 2022 HTHH eruption, the most
prominent signature was the Lamb waves which were globally observed by ground-based and spaceborne
geophysical instrumentations (Kulichkov et al., 2022; Liu et al., 2022; Lin et al., 2022; Matoza et al.,
2022; Themens et al., 2022; Adam, 2022; Kubota et al., 2022). Interestingly, we notice that a wide range
of the velocities from 280 m/s to 340 m/s were proposed through observations and Lamb wave modelling
(e.g., Kubota et al., 2022; Lin et al., 2022; Matoza et al., 2022; Themens et al., 2022). The travelling
velocity of Lamb waves in real atmosphere is affected by temperature distributions, winds and dissipation


(Otsuka, 2022). To investigate whether the propagation speeds of the lamb wave change in space and
time, we analyze the waveforms recorded by the DART buoys in the Pacific Ocean. The Pacific DART
buoys recorded the most discernible air-sea coupling pulse in deep ocean with Lamb waves that arrived
earlier than the theoretical tsunamis (Figure 1). The tsunami waveforms recorded by tide gauges did not
clearly detect the tsunami signals associated with the lamb waves, therefore not sufficient for further
analysis (Figure 2). Thus, we estimate the speed of Lamb waves using the waveforms recorded by the
Pacific DART buoys. The Lamb wave arrivals are limited within arrival time range from possible
velocities of 280–340 m/s. The time points at which the tsunami amplitudes first exceed 1 e-4 m above
sea level are defined as Lamb wave arrivals. By carefully fitting the arrivals with different constant
velocities, we illustrate the velocities of Lamb wave were generally uniform, but slightly decrease with
the increase of propagation distance (Figure 3). The Lamb waves initially propagated radially at speed
of ~340 m/s before slowing to ~325 m/s after reaching ~3400 km, and further decreasing to ~315 m/s at
7400 km. In an isothermal troposphere assumption, the phase velocity of the Lamb wave ($C_L$) can be
estimated with the following equation (Gossard and Hooke, 1975):
$$C_L = \sqrt{\frac{\gamma.R.T}{M}} \tag{2}$$
Where $\gamma$ =1.4 (air specific heat ratio corresponding to atmospheric temperature), R = 8314.36 J kmol-1
K-1 (the universal gas constant), M = 28.966 kg kmol-1 (molecular mass for dry air) are constant for the
air, T is the absolute temperature in kelvin. Thus, Lamb wave velocity is mainly affected by the air
temperature, meaning the travelling velocity of lamb waves might decrease when propagating from
regions with high temperature towards those with low temperatures, e.g., the north pole. By assuming a
set of possible temperatures in January (Table 1), we calculated the velocities CL could range between
312–343 m/s when temperatures vary between -30–20 °C. Therefore, the decreased velocity of the lamb
waves could be a consequence of cooling of the air temperature.

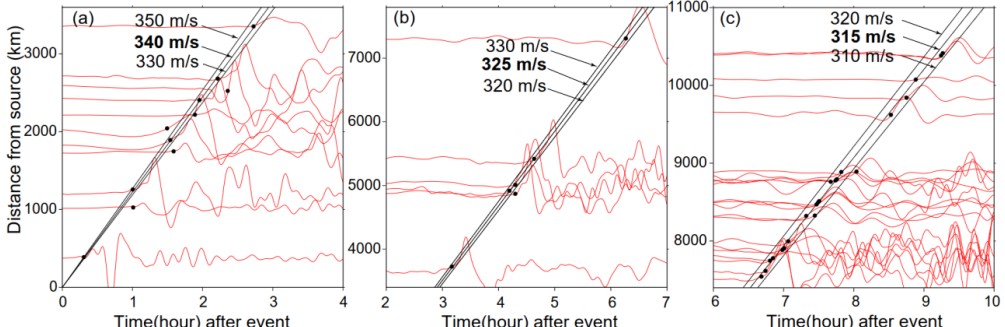

**Figure 3. Fitting the arrival times of normalized Lamb waveforms with different velocities. Black dots mark**
**the arrival times of the Lamb waves. Black lines represent velocities.**
**Table 1. Estimated Lamb wave velocities in an isothermal troposphere assumption**

| Celsius temperature (°C) | thermodynamic temperature (K) | $C_L$ (m/s) |
|---|---|---|
| 20 | 293.15 | 343.14 |
| 10 | 283.15 | 337.23 |
| 0 | 273.15 | 331.21 |
| -10 | 263.15 | 325.19 |
| -20 | 253.15 | 318.86 |
| -30 | 243.15 | 312.49 |

**3.2  Tsunami features observed by DART buoys and Tide gauges**
The statistics of tsunami heights and arrival times recorded at 38 DART buoys and 116 tide gauges across
the Pacific Ocean are used to interpret the tsunami characteristics. The comparison of the statistical
characters between DART and tide gauge observations yields some useful information of the
hydrodynamic process of tsunami propagation and help identify tsunami wave components with different
traveling velocities.
The average value of the maximum tsunami wave height (trough-to-crest) for the 116 tide gauge stations
is ~1.2 m. Figure 4a shows tide gauges with large tsunami heights exceeding 2 m are mainly distributed
in coastlines with complex geometries, such as gauges at New Zealand, Japan, and north and south
America. For example, the largest tsunami height among tide gauges is 3.6 m at a bay-shaped coastal
area Chañaral in Chile. In sharp contrast to tide gauges, the maximum tsunami heights of most Pacific
DART buoys are less than 0.2 m. The largest tsunami height in the DART buoys is only ~0.4 m recorded
at the nearest one, 375 km from the volcano (Figure 4b). The comparison between DART buoys and tide
gauges indicate that the direct contribution of air-sea coupling to the tsunami heights is probably in the





level of tens of centimeters (Kubota et al., 2022). The meter-scale tsunami heights at the coastlines
suggest the bathymetric effect could play a major role during tsunami propagation. In respect to the
arrival of maximum tsunami waves, the time lags between Lamb waves and the maximum heights of tide
gauges mainly range between ~0–10 h (Figure 4c). The delayed times of ~10 h are observed in New
Zealand, Hawaii, and west coast of America (Figure 4c), suggesting the interaction between tsunami
waves and local topography/bathymetry delay the arrival of the maximum waves (e.g., Hu et al., 2022).
The significant regional dependence of the coastal tsunami heights and the time lags of the maximum
tsunami waves can be attributed to the complexity of local bathymetry, such as continental shelves with
different slopes, and harbor/bay with different shapes and sizes (Satake et al., 2020). On the other hand,
since the DART records are less influenced by bathymetric variation in space, the first waves in DART
buoys are supposed to be the maximum tsunami waves as observed in the 2011 Tohoku tsunami event
(Heidarzadeh and Satake, 2013). However, we observe the inconsistency between the arrivals of the
Lamb waves and the maximum tsunami heights (Figure 4d). The time lags of the maximum waves of
DART buoys present a coarsely increasing tendency with the increasing distance from the volcano, which
indicates the contribution of other tsunami generation mechanism propagating with a uniform but lower
speed than Lamb wave.


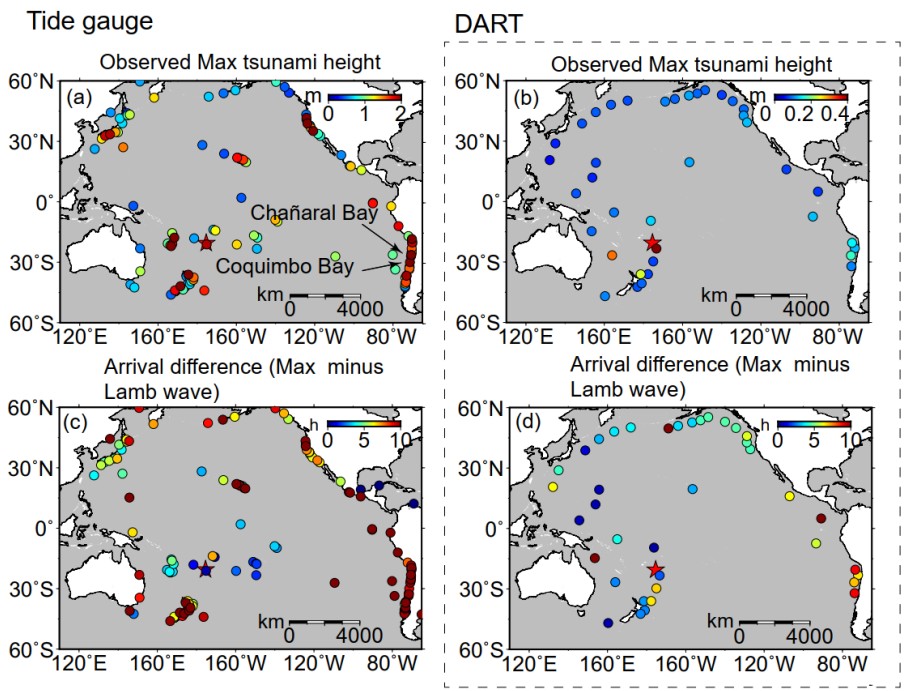


**Figure 4. The spatiotemporal signatures of the 2022 HTHH tsunami across the Pacific Ocean. (a) Observed**
**the maximum tsunami height (trough-to-crest height) of tide gauges. (c) Arrival differences between the**
**maximum tsunami height of tide gauges and Lamb waves. (b) and (d) are the same as (a) and (c) but for**
**DART buoys.**

**3.3 Tsunami components identified from wavelet analysis**

The statistical analysis of tsunami waveforms at tide gauges and DART buoys suggest the tsunami waves

likely contain several components with different source origins. To further identify these tsunami

components, we conduct wavelet analysis for tsunami waveforms recorded by representative DART

buoys and air pressure waveforms recorded by selected barometers. We demonstrate the analysis result

through the frequency-time (f-t) plot of wavelet which shows how energy and period vary at frequency

and time bands (Figure 5 and Figure 6). Tsunami components have clear signatures in all f-t plots as the

energy levels are quite large when they arrive. Figure 5 shows the wavelet analysis of six DART buoys

located in the vicinity of the eruption site (<3664 km). Figure 6 show the wavelet analysis of ten DART

buoys located in the Pacific rim which are far away from the source location. We observe three interesting

phenomena: 1) most of the tsunami wave energy is concentrated in four major period bands, i.e., ~80–

100 min, 10–30 min, 30–40 min, and 3–5 min; 2) The stations with 3-5 mins wave periods are mainly


located in the vicinity of the volcano site; 3) There exist one exceptional tsunami component with longer
wave period of ~80–100 min in the near source region which travels even faster than the lamb waves.
To further explore the source mechanism of these tsunami components, we take advantage of the
published information related to different propagating velocities of atmospheric gravity waves (Kubota
et al., 2022) and add four kinds of propagating velocities as criteria to differentiate the tsunami arrivals
from different sources (Figure 5 and Figure 6). The first reference speed is 1000 m/s related to the
radically propagating atmospheric shock waves near the source region (Matoza et al., 2022; Themens et
al., 2022). The second one is the velocities of Lamb wave ranging between 315–340 m/s derived from
the aforementioned analysis in section 3.1 (Figure 3). The third one is 200 m/s corresponding to the lower
limit of atmospheric gravity wave modes other than Lamb waves which were also excited by the volcanic
eruptions (Kubota et al., 2022). The last is the arrival time of conventional tsunami given by tsunami
modelling (Figure 1). The theoretical velocity of conventional tsunami is significantly nonuniform
spatially as compared with those of the atmospheric waves. The conventional tsunami propagation speed
is determined by the water depth along the propagation route. The velocity of non-dispersion shallow-
water waves ($C_H$) in the ocean is given by:
$$C_H = \sqrt{g.H}$$   (3)
Where g is gravity acceleration (9.81m/s$^2$), H is the water depth. The propagation velocities of tsunami
are ~296–328 m/s in the deepest trenches on earth (i.e., ~11 km in Mariana Trench and ~9 km in Tonga
Trench). The velocities decrease quickly to only ~44 m/s at ~200 m depth along the edge of continental
shelf. With the average depth of ~4–5 km, the average velocities in the Pacific Ocean range between
~198–221 m/s. Thus, theoretical tsunami velocities present significant slowness and variability. We
delineate the arrival times of the four reference speeds in Figures 5 and 6. For each panel of the figures,
from left to right, the solid vertical white lines mark velocity of 1000 m/s. The solid vertical red lines
mark the arrival of Lamb waves. The dashed vertical white lines mark lower limit of gravity waves'
velocity of 200 m/s. The dashed vertical black lines represent the calculated theoretical tsunami arrivals.
Horizontal white dashed lines mark two reference periods of 10 min and 30 min.
One particularly remarkable phenomenon is that the wave component with period of ~80–100 min
propagated at a very fast speed of ~1000 m/s in the vicinity of the HTHH site, i.e., New Zealand and
Hawaii (e.g., stations 52406, NZJ, NZE, 51425 in Figure 5). We infer that the tsunami component within




~80–100 min period band was likely produced by the atmospheric shock waves during the initial stage
of the volcanic eruption and spatially only cover the near-source region. To verify this observation, we
select 16 representative barometers located in the near-source region and far-field area for wavelet
analysis (see the locations in Figure 5 and Figure 6). Figure 7 shows the waveforms of atmospheric
pressure at selected locations and Figure 8 provides the frequency-time (f-t) plot of wavelet analysis of
some representative barometers. Interestingly, we are able to discern the air pressure pulses prior to Lamb
waves at barometers in New Zealand (the two columns on the left in Figure 7), although such signals are
not detectable in waveforms recorded by barometers far from the source (the two columns on the right
in Figure 7). The spatial distribution of such unusual pressure changes suggest that the fast travelling
shock waves were only limited in the near-source region, as reflected in the travelling ionospheric
disturbances (Matoza et al., 2022; Themens et al., 2022). Additionally, we also see that the long period
signals of ~80–100 min appear in DART buoys far away from the eruption site. Such signals may be
related with the long-period gravity waves (Matoza et al., 2022).
The tsunami components at period band of ~30–40 min can be readily associated with Lamb waves
because the arrival times of tsunami waves and Lamb waves have excellent match, as shown in the
tsunami data recorded by DART buoys (e.g., NZJ and 51425 in Figure 5; 51407, 32401 and 32413 in
Figure 6) and pressure data by barometers (Figure 8).
For the tsunami components with the period band of ~10–30 min, although the arrivals of ~10–30 min
tsunami components cover some theoretical tsunami arrival times, they do not consistently match. The
tsunami components occurring within the time period between Lamb waves and the lower gravity waves'
velocities has a good agreement with the velocity range of several atmospheric gravity wave modes
(Matoza et al., 2022; Themens et al., 2022; Kubota et al., 2022). Similarly, the air pressure data also show
energy peaks at ~10–30 min period band, which is consistent with the tsunami data (Figure 8). Such
consistency further verifies the contribution of atmospheric gravity waves to the volcanic tsunami.
The tsunami components with the shortest period of ~3–5 min (stations NZE, NZF, NZG and NZJ;
marked with black dashed squares in Figure 5) are only observed at DART records near the eruption
location. Meanwhile, the arrival times of these components agree well with the modelled arrivals of
conventional tsunami. Thus, we believe the observed shortest period band should originate from the
seafloor crustal deformation. We further infer that this component could be generated by the partial



underwater caldera collapse and/or subaerial/submarine landslide failures associated with 2022 HTHH
volcanic eruption.

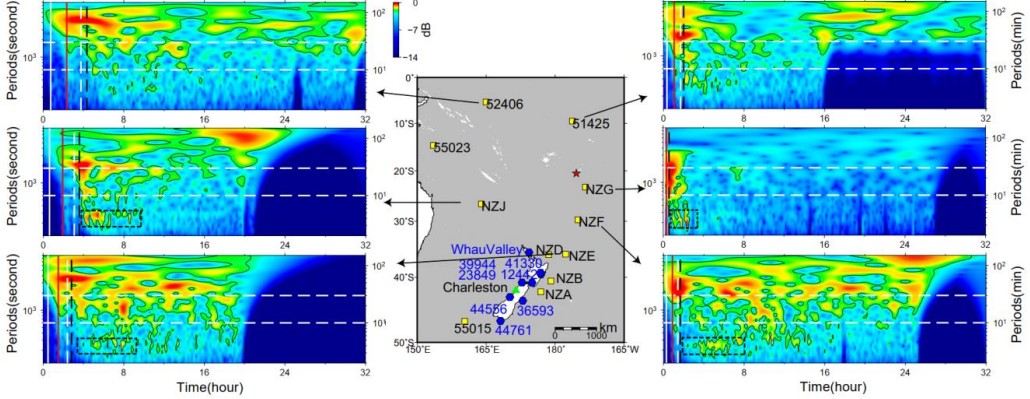


**Figure 5. Wavelet analysis of representative DART buoys in the vicinity of the HTHH volcano. In each sub-**
**plot, the solid vertical white lines mark the arrival time with travelling velocity of 1000 m/s. The solid vertical**
**red lines mark the arrivals of Lamb waves. The dashed vertical white lines mark lower limit of AGWs' velocity**
**of 200 m/s (Kubota et al., 2022). The dashed vertical black lines represent the theoretical tsunami arrivals.**
**The dashed horizontal white lines mark two reference wave periods of 10 min and 30 min. The blue hexagons**
**represent the locations of barometers. Green triangle makes the location of the tide gauges at Charleston.**


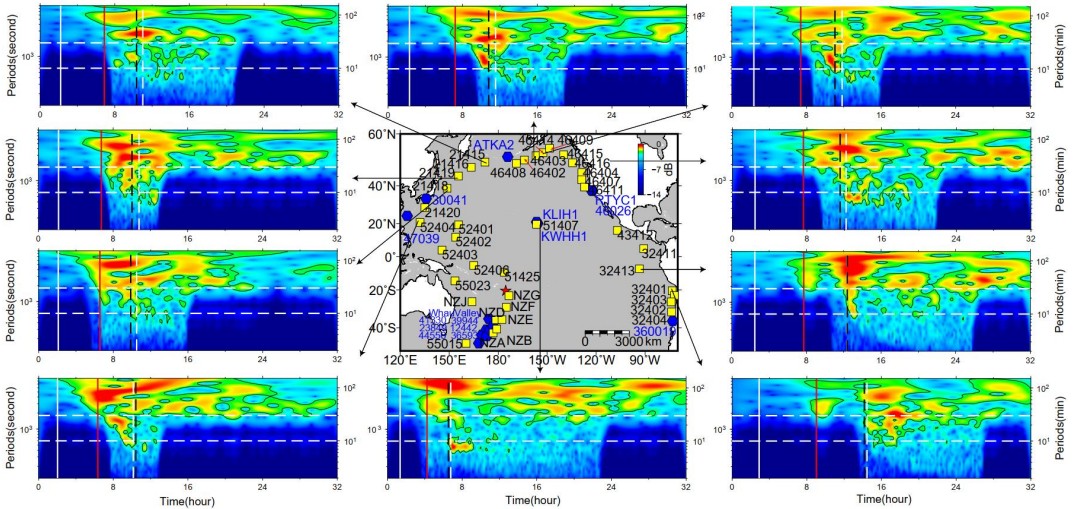


**Figure 6. Wavelet analysis of representative DART buoys far away from the HTHH volcano. In each sub-plot,**
**the solid vertical white lines mark the arrival time with travelling velocity of 1000 m/s. The solid vertical red**
**lines mark the arrivals of Lamb waves. The dashed vertical white lines mark lower limit of AGWs' velocity**
**of 200 m/s. The dashed vertical black lines represent the theoretical tsunami arrivals. The dashed horizontal**
**white lines mark two reference wave periods of 10 min and 30 min. The blue hexagons represent the locations**
**of barometers.**


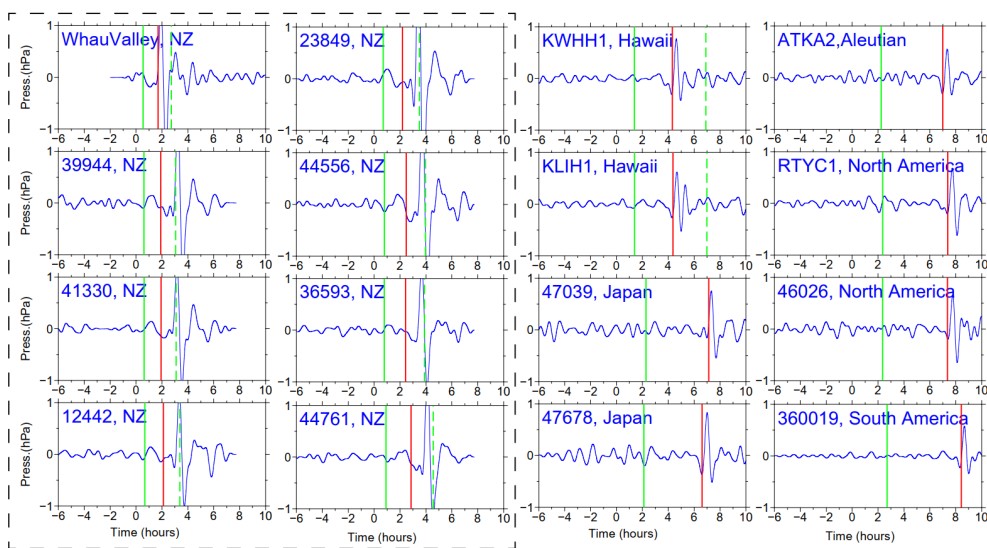

**Figure 7. Shockwave-related atmospheric pressure waveforms of selected barometers in the Pacific Ocean.
All traces have been filtered between 30 min and 150 min. In each sub-plot, the solid vertical green lines mark
the arrival time with travelling velocity of 1000 m/s. The solid vertical red lines mark the arrivals of Lamb
waves. The dashed vertical green lines mark lower limit of AGWs' velocity of 200 m/s.**

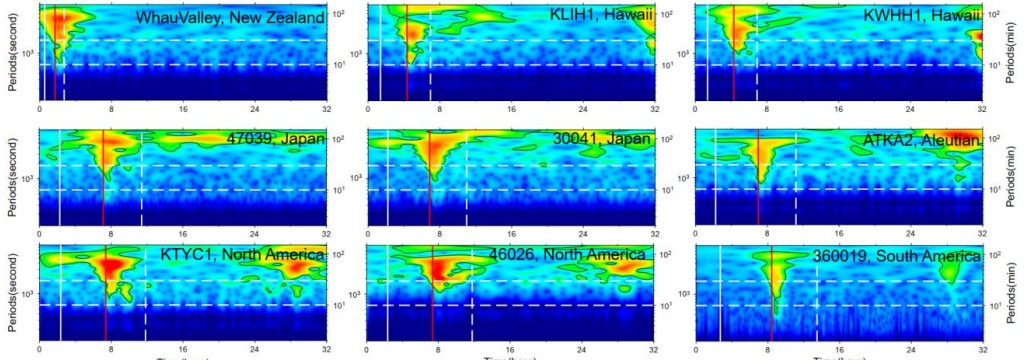

**Figure 8. Wavelet analysis of some representative barometers. In each sub-plot, the solid vertical white lines
mark the arrival time with travelling velocity of 1000 m/s. The solid vertical red lines mark the arrivals of
Lamb waves. The dashed vertical white lines mark lower limit of AGWs' velocity 200 m/s. The dashed
horizontal white lines mark three reference periods of 10 min and 30 min.**

## 4. Discussion

### 4.1 Tsunami from Caldera Collapse and Its Long-distance Traveling Capability

The tsunami wave energy distributed in different period bands is identified with reference arrival times.


The tsunami component with 3–5 min period is most likely generated by seafloor crustal deformation in
the volcanic site, but specific mechanism is not determined. A variety of possible scenarios associated
with the eruption could be responsible for the near-field tsunami waves, such as volcanic earthquakes,
pyroclastic flows entering the sea, underwater caldera flank collapse, and subaerial/submarine failures
(Self and Rampino, 1981; Pelinovsky et al., 2005). To further investigate the source mechanism, we
apply a simplified model to estimate the probable dimension of tsunami source:
$L = \frac{T\sqrt{gH}}{2}$                                                                              (4)
Where $L$ is the typical dimension (length or width) of the tsunami source, H is average water depth in the
source area, $g$ is the gravity acceleration, and T is primary tsunami period. By comparing with the post-
2015 morphology of the HTHH caldera which was obtained through drone photogrammetry and
multibeam sounder surveys, Stern et al. (2022) estimate that much of the newly-formed Hunga Tonga
Island and the 2014/2015 cone were destroyed by the 2022 eruption, and the vertical deformation of
Hunga Ha'apai Island is ~10–15 m (Stern et al., 2022). With no more quantitative constraint of the
seafloor deformation, we tentatively assume $H$ as 10–15 m, then the possible dimension of seafloor
crustal deformation responsible for the small-scale tsunami could be in the scale of 0.8–1.8 km (Figure
9a). The estimated size is very likely from partial caldera collapse that usually has limited scale in
volcanic site (Ramalho et al., 2015; Omira et al., 2022). If it is the case, the partial flank collapse could
be located between Hunga Tonga and Hunga Ha'apai Islands.

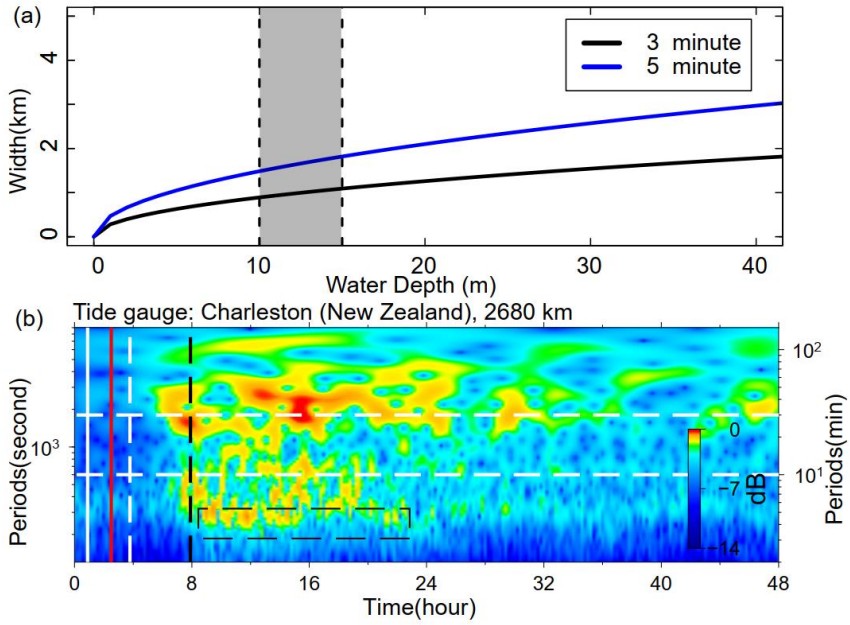

**Figure 9. Mechanism of tsunami component with 3–5min period. (a) The source dimension estimated by equation 4. (b) Wavelet analysis of tide gauge at Charleston, New Zealand, 2680 km away from the eruption site. The solid vertical white line marks the arrival time with travelling velocity of 1000 m/s. The solid vertical red line marks the arrival of Lamb wave. The dashed vertical white line marks lower limit of AGWs' velocity 200 m/s. The dashed vertical black line marks the theoretical tsunami arrivals.**

An interesting phenomenon is that the tsunami component with 3–5 min period can still be observed in a bay-shaped coastal area at Charleston in New Zealand (see the location in Figure 5) which is 2680 km away from the eruption site and maintains a high energy level lasting up to 14 h (Figure 9b). The long-traveling capability could be associated with the ~ 10000 m deep water depth of the Tonga Trench that keeps the source signals from substantial attenuation. In deep open ocean, the wavelength of a tsunami can reach two hundred kilometers, but the height of the tsunami may be only a few centimeters. Tsunami waves in the deep ocean can travel thousands of kilometers at high speeds, meanwhile losing very little energy in the process. The long oscillation can be attributed to the multiple reflections of the incoming waves trapped in the shallow-water bay at Charleston.

Generally, devasting tsunamis with long-distance travelling capability are mostly generated by megathrust earthquakes (Titov et al., 2005). Caldera collapses or submarine landslides with limited scale normally only generate local tsunamis, e.g., the 1998 PNG (Papua New Guinea) tsunami event (Kawata



et al., 1999) and the 1930 Cabo Girão tsunami event (Ramalho et al., 2015). Therefore, it's exceptional
that the tsunami component from scale-limited failure could travel at-least 2680 km away from the
eruption site. It demonstrates that tsunamis from small-scale tsunamigenic source have the capability to
travel long distance and cause long oscillation at favored condition, e.g., deep trench, ocean ridge and
bay-shaped coasts.
**4.2 The Possible Mechanisms of Long Tsunami Oscillation**
An important tsunami behavior of the 2022 HTHH tsunami is the long-lasting oscillation ~ 3 days in the
Pacific Ocean (Figure 10a), which is comparable to that of the 2011 Tohoku tsunami, ~4 days
(Heidarzadeh and Satake, 2013). We demonstrate the duration time of the tsunami oscillation through
ARMS (Averaged-Root-Mean-Square) approach that is a measure of absolute average tsunami amplitude
in a time period. The long-lasting tsunami energy can be observed at many regions, such as the coasts of
New Zealand, Japan, Aleutian, Chile, Hawaii, and west coasts of America. Several mechanisms could
account for the long-lasting tsunami, including (1) Lamb waves circling the Earth multiple times
(Amores et al., 2022; Matoza et al., 2022), (2) resonance effect between ocean waves and atmospheric
waves (Kubota et al., 2022), and (3) bathymetric effect. We discuss the contribution of each mechanism
in the following section.
To investigate the contribution of Lamb wave to the long-lasting tsunami, we compare the air pressure
disturbances recorded by selected barometers together with the tsunami waveforms of nearby tide gauges
(Figure 10b). While the barometers present discernible wave pulses at each Lamb wave's arrival, only
the first Lamb wave triggered clear tsunami signal and no detectable tsunami signatures correspond to
the following passage, suggesting the Lamb waves do not directly contribute to the long oscillation.
Theoretically, the resonance effects between ocean waves and atmospheric waves could contribute to the
long oscillation on coastlines based on the following reasons. First, part of the atmospheric gravity waves
propagated at velocities close to averaged velocities of conventional tsunami in the Pacific Ocean (198–
221 m/s) which resulted in the resonance with ocean waves (Kubota et al., 2022). Second, in deep oceanic
trenches, such as Mariana and Tonga-Kermadec trench (10000–11000 m), tsunami velocities range
between ~314–330m/s which are comparable with those of the observed Lamb waves 315–340 m/s.
When Lamb wave speed approaches the tsunami speed, Proudman resonance gradually increase tsunami
heights, wherein Proudman resonance optimally maximizes tsunami heights when they match well



(Tanioka et al., 2022; Lynett et al., 2022). Therefore, the resonance effect continuously supplied wave
energy to the ocean, especially in the deep trenches.
To examine the role of local bathymetry in the long-lasting tsunami, we choose a well-studied and well-
recorded event: the 2011 Mw 9.0 Tohoku tsunami as a reference event and compare the tsunami records
of these two events at the same coastal stations. Although the two tsunami events were generated by
completely different mechanisms, i.e., large-scale seafloor deformation for the Mw 9.0 megathrust
earthquake (Mori et al., 2011) and fast-moving atmospheric waves for the Mw 5.8 volcanic eruption
(Titov et al., 2005), they both produced widespread transoceanic tsunamis which were well recorded in
the Pacific DART buoys and tide gauges. In the near-field, the 2011 Tohoku earthquake produced runup
up to 40 m at the Iwate Prefecture, ~70 km from the source (Tanioka et al., 2022), while the 2022 HTHH
tsunami produced only ~13 m runup in the near field from eyewitness accounts in Kanokupolu, 60 km
from the volcano (Lynett et al., 2022). However, in the far-field (>1000 km), we observe comparable
tsunami wave heights in certain coastal regions. Based on the tsunami records at 21 tide gauges
surrounding the Pacific Ocean, Heidarzadeh & Satake (2013) calculated the average value of the
maximum tsunami heights (trough-to-crest) of the 2011 Tohoku tsunami is 1.6 m with the largest height
of 3.9 m at the Coquimbo Bay in Chile (Heidarzadeh and Satake, 2013). Coincidently, the statistics of
116 tide gauges in this study also suggest the average tsunami heights of the 2022 HTHH tsunami is
around the same order, ~1.2 m, among which, the largest height is 3.6 m at Chañaral Bay in Chile.
Interestingly, in the coastal region of South America, the locations of the largest tsunami heights of both
events are adjacent (Figure 4a), i.e., Coquimbo (the 2011 Tohoku) and Chañaral (The 2022 HTHH).
To further compare the far-field hydrodynamic processes between these two events quantitatively, we
conduct wavelet analysis for four representative tide gauges distributed across the Pacific Ocean, i.e.
coastal gauges at East Cape in New Zealand, Kwajalein Island, Wake Island, and Talcahuaho in Chile
(see their locations in Figures 10b). The temporal changes of tsunami energy of both events can be seen
in Figure 11. At each tide gauge, the tsunami energy of the 2011 HTHH (Figure 11a) and the 2022 Tohoku
tsunamis (Figure 11b) for the first few hours after the arrivals is nonuniform with different significant
peaks distributed within a wide period band of ~3–100 min. Then, the following long-lasting energy of
the both at each station presents similar pattern and is concentrated at identical and fairly narrower period
channel, i.e., ~20–30 min at East Cape in New Zealand, ~40–60 min at Kwajalein Island, ~10 min at


Wake Island, and ~100 min at Talcahuaho in Chile, which reflects the local bathymetric effects of natural
permanent oscillations (Hu et al., 2022; Satake et al., 2020). Specifically, many bathymetric effects can
contribute to the long-lasting tsunami, such as multiple reflections across the basins, or the continental
shelves, and the excited tsunami resonance in bays/harbors with variable shapes and sizes (Aranguiz et
al., 2019; Satake et al., 2020). For example, tide gauges around New Zealand are primarily distributed in
harbors/ports with major natural oscillation modes of ~20–30 min (De Lange and Healy, 1986; Lynett et
al., 2022). The first oscillation mode of central Chile is centered around ~100 min (Aranguiz et al., 2019).
Consequently, Figure 11 illustrates that the long-lasting tsunami energy of the two events is respectively
distributed in 20–30 min period at East Cape in New Zealand and in ~100 min period at Talcahuaho in
central Chile. The coupling of bathymetric oscillation mode with tsunami containing similar-period wave
results in the excitement of tsunami resonance, which amplifies tsunami waves and prolongs the tsunami
oscillation at the two stations (Heidarzadeh et al., 2019, 2021; Hu et al., 2022; Wang et al., 2022).
Simply put, atmospheric acoustic-gravity waves from the 2022 HTHH eruption do not directly contribute
to the long-lasting tsunami, but the resonance effect associated with ocean waves theoretically could
contribute to it. However, the similarity of far-filed hydrodynamic behaviors between the 2022 HTHH
volcanic tsunami and the 2011 Tohoku seismogenic tsunami demonstrates the both went through similar
hydrodynamic processes after their arrivals. The consistency favors that the long-lasting tsunami of 2022
HTHH tsunami event can very likely be attributed by the interplays between local bathymetry and
conventional tsunami left after each passage of atmospheric waves, which can well explain why the two
completely distinct tsunami events possess a comparable duration time.
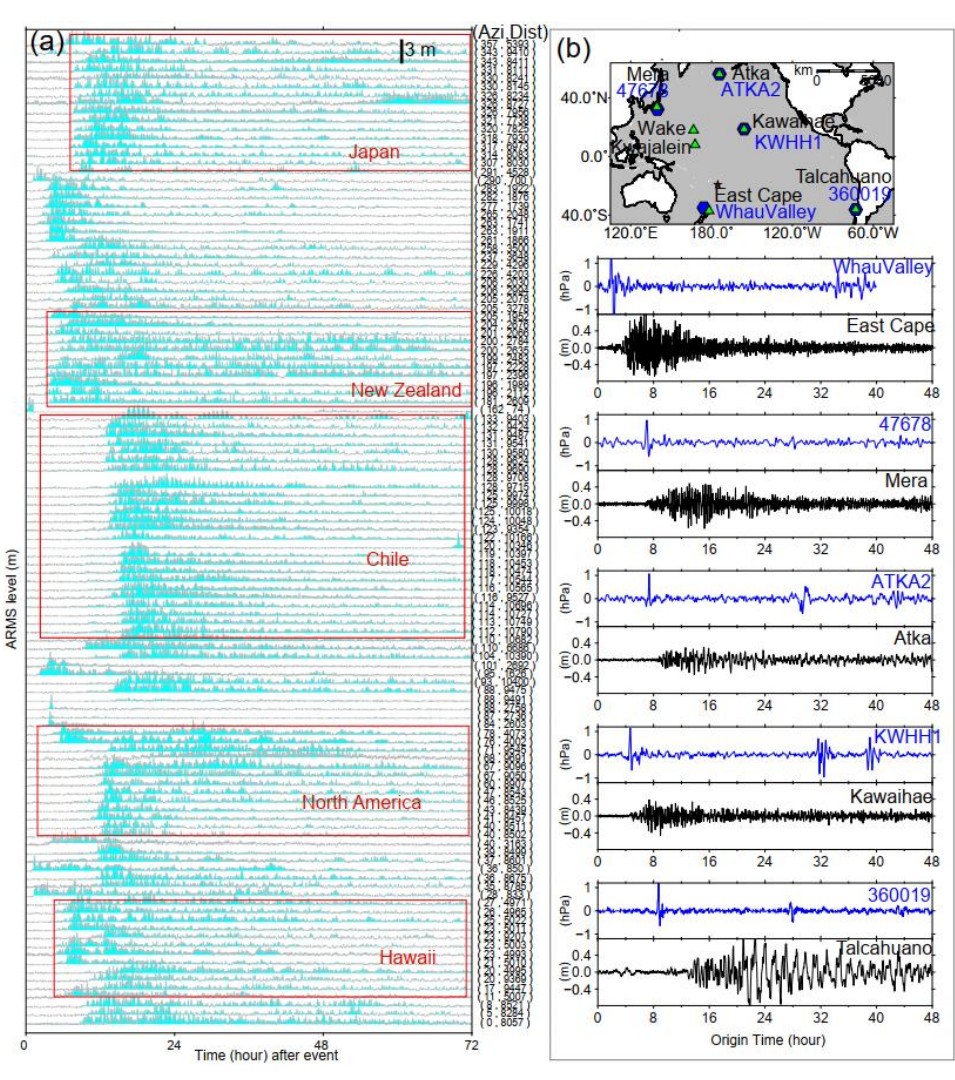

**Figure 10. Tsunami duration. (a) Tsunami durations at Pacific 116 tide gauges through ARMS level approach.**

**(b) the location of barographs (blue curves) and nearby tide gauges (green curves), as well as their waveforms.**

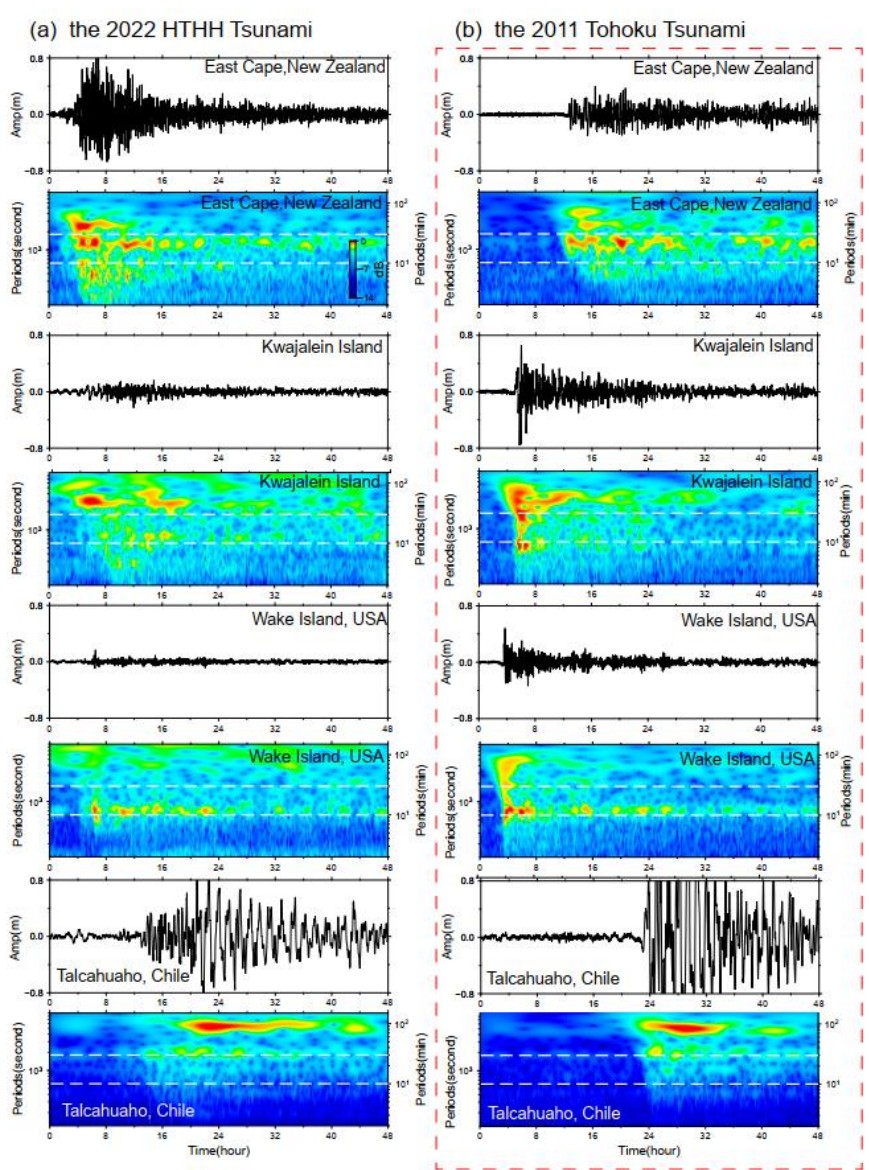

**Figure 11. Wavelet analysis of tsunami waveforms recorded by 4 tide gauges during (a) the 2022 HTHH tsunami event, and (b) the 2011 Tohoku tsunami event.**

**4.3 Challenges for Tsunami Warning**

The generation mechanisms and hydrodynamic characteristics of the 2022 HTHH volcanic tsunami are more complicated than pure seismogenic tsunami, which challenge the traditional tsunami warning approach.

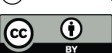

The first challenge is posed by the tsunami components with propagating velocities faster than the
conventional tsunami. The Tonga volcanic tsunami event provides an excellent example which highlights
that the tsunamigenic mechanisms are not limited to tectonic activities related with the sudden seafloor
displacements, but also include a variety of atmospheric waves with distinct propagation velocities. The
tsunami components in 2022 HTHH event generated by the air-sea coupling possess a wide range of
velocities from 1000 m/s to 200 m/s. The Lamb waves recorded in both the 2022 HTHH event and the
1833 Krakatoa volcanic event traveled along the Earth's surface globally for several times (Carvajal et
al., 2022). The tsunami waves produced by Lamb waves, the wave components associated with resonance
of the air-sea coupling and their superimposition increase the difficulty of tsunami warning.
Another critical challenge is associated with the interplays between tsunami waves and local bathymetry.
The tsunami waves left by each passage of the atmospheric waves can interact with local bathymetry at
coastlines, such as continental shelves with different slopes, and harbor/bay with different shapes and
sizes. The interaction can intensify the tsunami impact and excite a variety of natural oscillation periods.
The 2022 HTHH tsunami with an extremely wide period range of ~2–100 min have a great potential to
couple with the excited natural oscillations and form extensive tsunami resonance phenomena. The
resonance effects result in long-lasting oscillation and delayed tsunami wave peaks. The uncertain
arrivals of the maximum tsunami waves pose an extra challenge to tsunami warning.
**5.  Conclusion**
In the study, we explore the tsunamigenic mechanisms and the hydrodynamic characteristics of the 2022
HTHH volcanic tsunami event. Through extensive analysis of waveforms recorded by the DART buoys,
tide gauges and barometers in the Pacific Ocean, we reach the main findings as follows:
(1) We identify four distinct tsunami wave components based on their distinct propagation velocities or
period bands (~80–100 min, 10–30 min, 30–40 min, and 3–5 min). The generation mechanisms of these
tsunami components range from air-sea coupling to seafloor crustal deformation during the volcanic
eruption.
(2) The first-arriving tsunami component with 80–100 min period was most likely from shock wave
spreading at a velocity of ~1000 m/s in the vicinity of the eruption. This tsunami component was not
clearly identified by currently available publication and it's not easy to be visually observed through time



series of the waveforms. The physical mechanism is yet to be understood. The second tsunami component
with 30–40 min period was from Lamb waves, and was the most discussed tsunami source of this event
so far. A thorough analysis of DART measurements indicates that the Lamb waves traveled at the speed
of ~340 m/s in the vicinity of the eruption and decreased to ~315 m/s when traveling away due to cooling
of the air temperature. The third tsunami component was from some atmospheric gravity wave modes
with propagation velocity faster than 200 m/s but slower than Lamb waves. The last tsunami component
with the shortest periods 3-5 min was probably produced by partial caldera collapse with estimated
dimension of ~0.8–1.8 km.
(3) The long-lasting Pacific oscillation of this tsunami event was not only associated with the resonance
effect with the atmospheric acoustic-gravity waves, but more importantly the interactions with local
bathymetry. The velocities of tsunami waves in deep ocean (especially at Mariana and Tonga-Kermadec
trenches) close to those of acoustic Lamb waves and some gravity wave modes produced resonance
effects, which supplied energy to the ocean. The comparison of hydrodynamical characteristics between
the 2022 HTHH tsunami event and the 2011 Tohoku tsunami event suggests the volcanic tsunami
oscillation was prolonged by their interplays with local bathymetry.
(4) The extraordinary features of this rare volcanic tsunami event challenge the current tsunami warning
system which is mainly designed for seismogenic tsunamis. It is necessary to improve the awareness of
people at risks about the potential tsunami hazards associated with volcanic eruptions. New approaches
are expected to be developed for tsunami hazard assessments with these unusual sources: various
atmospheric waves radiated by volcanic eruptions besides those traditionally recognized, e.g.
earthquakes, landslides, caldera collapses and pyroclastic flows etc.
**Acknowledgment**
This work was supported by National Natural Science Foundation (No 41976197, No 12002099),
Innovation Group Project of Southern Marine Science and Engineering Guangdong Laboratory (Zhuhai)
(No. 311021002), Key Research and Development Program of Hainan Province (No. ZDYF2020209),
Southern Marine Science and Engineering Guangdong Laboratory (Zhuhai) (SML2021SP305) and
Fundamental Research Funds for the Central Universities, Sun Yat-sen University (2021qntd23).
The JAGURS tsunami simulation code is employed for tsunami modelling (Baba et al., 2015;



https://zenodo.org/record/6118212#.Yk98qdtBxPY). Bathymetry data are obtained from GEBCO at
http://www.gebco.net. The sea level records in deep ocean are available from the Deep Ocean Assessment
and Reporting of Tsunamis (DART) buoy network in the Pacific (https://nctr.pmel.noaa.gov/Dart/), and
GeoNet New Zealand DART network (https://tilde.geonet.org.nz). The sea level records of tide gauges
are downloaded from UNESCO/ IOC (http://www.ioc-sealevelmonitoring.org/). Barometer data are
provided    by    the    following    providers:    Direccio´n    Meteorolo´gica    de    Chile
(https://climatologia.meteochile.gob.cl),    NOAA    National    Weather    Service
(https://www.weather.gov/ilm/observations), Japan Meteorological Agency (https://www.jma.go.jp),
The UK Met Office Weather Observation (https://wow.metoffice.gov.uk/observations), and Fiji
Meteorological Service (https://www.met.gov.fj).

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
