# Peer review of "The characteristics of the 2022 Tonga volcanic tsunami in the Pacific Ocean"

_Natural Hazards and Earth System Sciences, 2022_

## Author Response (AR1)

October 10th, 2022

Dear Dr. González,

We sincerely thank you and two reviewers for the constructive comments that greatly helped us to improve the manuscript. Here we present our point-by-point responses and revisions to the comments.

Linlin Li

Note: The comments are in "*italics*", and our responses and revisions are in "regular" text (in blue) for clarity.

━━━━━━━━━━━━━━━━━━━━━━━━━━━━━━━━━━━━━━━━━━━━━━━━━━━━━━━━

**Response to Reviewer #1**

**General Comments:**

*This is a well-written paper. The authors provide a comprehensive summary of the present studies of the 2022 Tonga tsunami, and also present their research results derived from careful analysis of massive records of tsunami data. The suggest that the tsunami data consist of four different components according to their periods due to different mechanisms. I think the current form is acceptable, except a few minor clarifications and editorial changes.*

**Author's response:** We thank referee #1 for the positive comments and encouragements he/she made on the manuscript. Here, we present our point-by-point responses and revision to each of the comments, and believe that the manuscript will be improved as a result of these changes.

*2.2 Tsunami Modeling:*
*JAGURS can consider secondary effects on tsunami propagation, e.g., Earth elasticity and seawater stratification. Did you include these effects in your simulations?*

**Author's response:** For the secondary effects, we have considered the dispersion effect which is important for tsunamis generated by non-seismic sources (e. g. flank failure or landslides caused seafloor crustal displacements). As the purpose of using tsunami modelling approach is to obtain the arrival times of conventional tsunami, we only take into account the secondary effect which has appreciable influence on tsunami arrivals. Among the impact of secondary effects on modelled arrival times, the dispersion effect has a relatively bigger influence in the arrival times in the far-field (Tsai et al., 2013; Watada, 2013). The discrepancy between arrival times including and not including the dispersion effect is ~ ten min. Since the Earth elasticity and seawater stratification have very minor influence on tsunami arrivals, they are not considered in the modeling of the 2022 Tonga volcano tsunami event.

*Somewhere in the text the authors may emphasize that the DART data are actually pressure records, instead of direct water height. Thus, these records can be real pressure in Pascals if the signals are shock or Lamb waves. This is different as the coastal gauges are only water height.*

**Author's response:** We thank referee #1 for pointing out this important information. We've added the sentences in the manuscript based on the suggestion

**Author's change to manuscript:** We've added a sentence in the related context: "DART buoy with pressure sensor deployed at the ocean's bottom records the sea level change that is transferred from pressure records in Pascals, instead of direct water height. For the 2022 HTHH tsunami event, the pressure fluctuation at DART buoy is a superposition of the pressure changes caused by tsunami and the Lamb wave (Kubota et al., 2022)."

*Line 181: CL -> L:subscript*
*Line 207: delay -> delays*
*Line 235: exist -> exists*

**Author's response:** Thanks. Done as suggested.

**Reference:**

Kubota, T., Saito, T., and Nishida, K.: Global fast-traveling tsunamis by atmospheric pressure waves on the 2022 Tonga eruption, Science (80-. )., https://doi.org/10.1126/science.abo4364, 2022.

Tsai, V. C., Ampuero, J. P., Kanamori, H., and Stevenson, D. J.: Estimating the effect of Earth elasticity and variable water density on tsunami speeds, Geophys. Res. Lett., 40, 492–496, https://doi.org/10.1002/grl.50147, 2013.

Watada, S.: Tsunami speed variations in density-stratified compressible global oceans, Geophys. Res. Lett., 40, 4001–4006, https://doi.org/10.1002/grl.50785, 2013.

═══════════════════════════════════════════════════════════

**Response to Reviewer #2**

**General Comments:**

*In this study, authors conducted wavelet analyses to investigate the characteristics of the 2022 Tonga volcanic tsunami, which is, to date, the most important event in the geoscience field in 2022. The MS presents some interesting results. Nevertheless, there are still many flaws, which need to be further polished, clarified, and validated with deep and serious thinking. My comments include,*

**Author's response:** We are very grateful to referee #2 for the detailed advices and comments which definitely are very helpful in improving the clarity and rigor of this

manuscript. Here, we present our point-by-point responses and revision to each of the comments.

**Author's change to manuscript:** Please see the detailed changes in our response to each comment.

*1.Line 21, Lamb wave with ~30-40 min period? So long?*

**Author's response:** The Lamb wave is well studied as the signature is clear and conspicuous in both sea surface and atmosphere. Its velocity of ~300-340 m/s is distinctly different from that of gravity waves of ~200-220 m/s (Kubota et al., 2022) and shock waves of ~1000 m/s. Therefore, we can well identify the arrival of Lamb wave from the waveforms recorded by DARTs and barometers. By carefully analyzing their wavelet, we find the period of Lamb wave is ~30–40 min. The 30-40 min period is consistent with the dominant periods 1700–2500 s (28.3–41.6 min) identified by Matoza et al., 2022 who did the analysis of a mass of barometric and seismic data. The period is also similar with the Lamb wave period of 2000 s (33.3 min) detected by 25 comprehensive infrasound sensors installed along the coastline of Japan with wider frequency range than conventional barometers (Nishikawa et al., 2022). Such a long period is probably excited by the long-duration climactic eruption (Matoza et al., 2022).

*2.Lines 107-108, Why using cut-off frequency of ~8 hr could remove the tidal components? There are various tidal components.*

**Author's response:** The dominant periods of astronomical tides are generally ~10 hr. The subperiods (Power relation of 0.5 with the dominant period) of the tide are not significant compared with the dominant one and can be mixed with the excited nontidal oscillation periods (Parker, 2007). For example, in shallow waterways, nontidal phenomena such as river flow and low-frequency storm surge can affect the amplitudes and phases of some tides. In the case, we use the cut-off frequency of ~8 hr to keep as much as tsunami information and meanwhile remove the dominant tide components. This frequency has been successfully applied to many tsunami cases. For example, it works well on tsunamis generated by the 2011 Mw 9.0 Tohoku megathrust earthquake (Heidarzadeh & Satake, 2013) and the 2022 Mw 6.9 normal-faulting intraplate earthquake (Doğan et al., 2021).

*3.Line 110, Why ignore such small tsunami height data?*

**Author's response:** We explain that we only delete the tidal gauge stations with the maximum tsunami height less than 0.2 m. All the DART stations remain as tsunami waves in deep ocean possess very small wave amplitudes. The reasons we apply this step to tide gauge data are: First, the gauges distributed along different locations in the Pacific coastline possess different scales of noise signal. To minimize the effect of the noise on the waveform analysis, we ignore stations whose tsunami height is close to that of noise signal. Second, data from the remaining tide stations are sufficient to demonstrate our key points.

**Author's change to manuscript**:For clarity, we've rephrased the sentence as : '…with the maximum tsunami heights of tide gauges less than 0.2 m…'.

*4.Please specify the azi definition in the figure caption, specify the unit of distance. Please also add the magnitude of the ordinate to quantitatively specify the sea level.*

**Author's response**:We've modified the part about "azi and distance unit" accordingly. The heights of the stations are distributed within a wide range from ~3 m to 0.2 m. To clearly demonstrate the data with such a wide range, we choose to normalize the data instead of using the raw data. That's the reason we didn't put a scale in the figure. The previous ordinate is misleading. We've corrected the label of the ordinate.

**Author's change to manuscript**:We've added the units in the figure 2 and a sentence to specify the Azi definition in the caption as: "Azi stands for azimuth". We've corrected the "sea level (m)" with "Normalized wave amplitude".

[Figure]

**Figure 2. Detided tsunami waveforms at (a) DART buoys and (b) tide gauges. Waveforms in both subplots are shown in ascending distance. Azi stands for azimuth.**

*5.Line 134, Why Gaussian-shaped initial sea level displacement is used? Can it mimic the volcano eruption induced gravity wave propagation?*

**Author's response:** The main purpose of using Gaussian-shaped point source is to calculate the theoretical arrival times of conventional tsunami generated by seafloor crustal displacement. The Gaussian-shaped point source probably cannot realistically mimic the volcano eruption induced gravity wave propagation, but the arrival times are not very sensitive to the source property in this specific case.

*6.Line 144, Why Morlet mother function is selected?*

**Author's response:** Morlet wavelets have several advantages for time-frequency analysis. First, it is Gaussian-shaped in the frequency domain. The absence of sharp edges minimizes ripple effects. Second, wavelet convolution is more computationally efficient, because most of which are implemented with the fast Fourier transform. Third, the convolution result can retain the temporal resolution of the used signal (Cohen, 2018).

*7.Line 149, Why moving time window is selected as 20 min?*

**Author's response:** We use moving time window in the Averaged-Root-Mean-Square (ARMS) method to measure the absolute average tsunami amplitude in the window for coastal tide gauge. To achieve this purpose, we need to choose a representative wave period that covers most tsunami waves with significant amplitude and long-lasting time duration, as the moving time window. With the criterion, we exclude the periods of tsunami components from shock wave, Lamb wave and conventional seafloor crustal displacement, because they are either distributed in a limited time period in each waveform, or have very limited spatial and temporal impact, therefore, do not meet the requirements. Specifically, significant shock wave and Lamb wave are mainly concentrated in the narrow time periods of their arrivals. Conventional tsunami is only observed in the proximity of the eruption site. Only tsunami component from the air-sea coupling with atmospheric gravity wave possesses significant waveform for a long time and the wave period band of such coupled waves is ~10-30 mins. Therefore, we choose 20 min as the moving time window. Base on our test, an alternative time window in the range of 10-30 min can also get the similar duration result.

*8.Lines 163-164, This could not be observed in Fig. 1. Please specify the theoretical (gravity wave) tsunami speed in Fig. 1 to show that Lamb wave is faster.*

**Author's response:** Thanks, done as suggested.

**Author's change to manuscript:** We've added the marks "Theoretical tsunami arrival" and "Lamb wave arrival" in Figure1.

[Figure]

*9.Lines 164-165, Why Fig. 2 could not detect the Lamb wave related tsunami signals?*

**Author's response:** Lamb waves is clear in waveforms recorded by DARTs (the left column) because DARTs deployed in deep ocean capture clear Lamb waves. However, tsunami left by each passage of the atmospheric waves in tide gauges (right column) is much affected and amplified by the complicated coastlines and local bathymetric features, which render the lamb wave not that conspicuous in tide stations, so it's harder to see clearly outstanding Lamb wave signatures in the coastal gauges.

*10.Line 165, Lamb, L should be capital. Please double check this throughout the entire MS.*

**Author's response:** Thanks, done as suggested.

*11.Lines 168-169, Why such definition? In data pretreatment, data with the maximum tsunami height less than 0.2 m have been deleted? Nevertheless, a very small value of 0.1 mm (could be recording error in many data) is considered here?*

**Author's response:** The definition is based on our tests, in which 0.1 mm can well represent the amplitude of Lamb wave arrivals, so the time points at which the tsunami amplitudes first exceed 0.1 mm above sea level are defined as Lamb wave arrivals. About the usage of 0.2 m, we only delete the tidal gauge stations with the maximum tsunami height less than 0.2 m. All the DART stations remain as tsunami waves in deep ocean possess very small wave amplitudes.

*12.Line 178, In Eq. 2, temperature is for low elevation or high elevation? If low, then moving towards North Pole is accompanied with the decease of temperature, thus the decreased Lamb wave speed. However, if it is temperature at high elevation, the above explanation fails.*

**Author's response:** The equation is built on an assumption of an isothermal troposphere, the phase velocity is only affected by the air temperature. DARTs we use measure the sea surface elevation in deep ocean, so the temperatures we obtain are for low elevation. The equation has been successfully applied in numerical simulation of atmospheric Lamb waves of 2022 Tonga eruption (Amores et al., 2022).

*13.Line 181, CL, L should be subscript.*

**Author's response:** Thanks, done as suggested.

*14.How the black lines are obtained? They are very much sensitive. Please add wave height elevation information in Fig. 3.*

**Author's response:** The black lines represent different constant velocities (in Fig. 3). We set the velocity to fit the Lamb wave arrival, in order to obtain the spatial variation in Lamb wave velocities. The waveforms are normalized instead of using the raw data, so it's may be confusing to add elevation information.

*15.Line 196, Can not confirm the complex geometries of the coastlines in Fig. 4a.*

**Author's response:** Thanks. We have added a figure of the coastlines of Japan (Figure S1a) in the supplementary as a representative, to present the complex geometries of the coastlines.

[Figure]

**Figure S1. Bathymetry in Japan (a) and Chañaral bay (b) in Chile. Bathymetry data is downloaded**

**from GEBCO (http://www.gebco.net ).**

*16.Line 198, Can not see the bay shape in Chanaral.*

**Author's response**:We have added a figure of the bay in Chanaral (Figure S1b) in the supplementary, to present the bay shape in Chanaral.

*17.Lines 206-207, Why interaction between tsunami and bathymetry could delay the arrival of maximum tsunamis? There are always interactions between bathymetry and tsunami propagation. It is inherent.*

**Author's response**:Tsunami interaction with different bathymetry features can lead to various effect. Some bathymetric effect can delay the arrival of maximum tsunamis. For example, the edge waves (Satake et al., 2020) and resonance effect (Wang et al., 2021) from tsunami interaction with different local bathymetry can produce late maximum tsunami amplitude (Satake et al., 2020). The interaction phenomenon between tsunami and bathymetry is better understood for conventional tsunami originated from seafloor crustal displacements, but it's not well studied for atmospheric tsunami from volcanic eruption as it's so rare and complicated. So, we feel it's necessary to emphasize the idea here. To make the delayed cause more clear, we added the specific phenomenon in the main context.

**Author's change to manuscript**:We added a sentence in the related context: "For example, the delayed maximum tsunami height can be attributed to the edge waves (Satake et al., 2020) and resonance effect (Wang et al., 2021) from tsunami interplays with bays/harbors, islands, and continental shelves of various sizes."

*18.Lines 211-214, Why the first waves in DART are supposed to be the maximum? The first wave is induced by the Lamb wave, it is small (should be only about 2 cm corresponding to 2 hPa), whereas the maximum waves should come from other mechanisms.*

**Author's response**:Since the DART stations are located in the deep ocean, the contribution from shoaling effect and interaction with complex coastlines is relatively limited. Based on previous observations (e.g. Heidarzadeh & Satake, 2013), the first tsunami waves are normally the largest waves at most DART stations. The delayed maximum waves suggest other mechanism might have contributed to the tsunami case, which has been proven to be atmospheric gravity waves.

*19.Lines 233-234, Why not in sequence? These bands cover almost all time period in Fig. 5.*

**Author's response**:Thanks, we have corrected the sequence.

**Author's change to manuscript**:We've rephrased the sentence as: "… 3–5 min, 10–30 min, 30–40 min, and ~80–100 min …"

*20.Lines 234-235, Please be specified.*

**Author's response:** Thanks, done as suggested.

**Author's change to manuscript:** We've rephrased the sentence as: "2) The significant tsunami component with period band of 3-5 mins are recorded by stations between the eruption site and the north tip of the New Zealand."

*21.Lines 235-236, Please be specified.*

**Author's response:** Thanks, done as suggested.

**Author's change to manuscript:** We've rephrased the sentence as: "3) There exist one exceptional tsunami component with longer wave period of ~80–100 min mainly recorded in the Tonga, the New Zealand and Hawaii, which travels even faster than the lamb waves."

*22.Lines 256-260, No need since these have be specified in the figure caption.*

**Author's response:** Thanks, removed as suggested.

*23.Line 263, There is no Hawaii in Fig. 5. In fact, the 80-100 min wave energy in these two regions on the left of the vertical white line is rather small, and no clear difference from other points.*

**Author's response:** We've added Hawaii station in the sentence. The 80-100 min wave energy is supposed to be around the vertical white lines instead of on the left of the lines. As all stations in Fig. 5 are located in New Zealand, close to the eruption site, the energy of the tsunami components occurs at the similar time and couple together, which therefore may be little hard to distinguish for some stations (such as NZG, NZF and NZJ). The small energy of the 80-100 min wave distributed in the stations NZG, NZF and NZJ could also attribute to the complicated air-sea coupling conditions, which is poorly understood because of limited observation. However, Fig. 7 shows that the clear and consistency of the component are recorded by barometers in New Zealand.

**Author's change to manuscript:** We have modified the text as : …e.g., stations 52406, NZJ, NZE, 51425 in Figure 5, and 51407 in Fig. 6

*24.why the signals are filtered between 30 min and 150 min, whereas the period band is ~80-100 min in line 261. Why different?*

**Author's response:** To clearly show the ~80–100 min wave component which arrives prior to Lamb wave in figure7, we have to keep Lamb wave period component as a reference so we start from 30 min (lower period limit of the Lamb wave). ~80–100 min is a general period band of the tsunami component, not an exact value. Periods of some tsunami components are longer than 100 min. To keep as much as period information of various tsunami components,

we choose 150 min as the upper period limit. According to our tests, changing the upper period to 130 min or 140 min does not affect the results.

*25.Lines 269-272, Hard to identify this in Fig. 7. There are no clear difference between the left and right two columns regarding the signals around the vertical solid green line.*

**Author's response:** Indeed, the signals are not visually conspicuous as the Lamb waves. But they are relatively well detected by the wavelet analysis of waveforms recorded by both DARTs and barometers in New Zealand. And compared with background noise signal prior to Lamb wave, the waveforms of the stations in New Zealand (For example, WhauVaully, 39944, 12442, 44556, 44761) have more clear amplitudes at the green lines than those in the rest of station in the Pacific Ocean.

*26.Lines 285-286, The large energy of the air pressure of 10-30 min band in Fig. 8 only appears around the arrival of Lamb wave, while large energy of tsunami wave of 10-30 min band shows a much longer duration in the volcano near field in Fig. 5 (after the arrival of Lamb wave), and a relatively short duration in the volcano far field in Fig. 6 (after the arrival of Lamb wave). I do not think Figs. 5, 6 and 8 are consistent with respect to this point. Appreciate if authors could further dig out the physical insights behind.*

**Author's response:** Thanks for raising such interesting question which encourage us to explore further of our results. By careful checking the original waveform data, we realize the duration variability shown in different DARTs is actually related to the duration of event mode set by each DART station. Passing tsunami waves trigger the DART system to enter event mode (with sampling rate of 15 seconds or 1 min) from normal frequency mode (15 min time interval) (www.ndbc.noaa.gov/dart). When the tsunami signals eventually die down, the recording frequency will be switched back to normal mode. The 10-30 min tsunami component can be well detected by the sampling rate of 15 second or 1 min, but not by the rate of 15 min. Therefore, we see different durations of 10-30 min band in Figures 5 and 6.

*27.The colorbar seems strange. It should represent the energy. Why negative values? what is the meaning of dB? As for the left and right ordinates of each sub-figure, their scales are different, left around 10^3, while right around 10^1. Why? As for the wavelet results, why there is no blanked-out peripheral area of the spectrum, i.e., 'cone of influence', the portion of the spectrum sensitive to the end-effects. These areas should be blanked where results may be artificially affected. Similar problems for the entire wavelet analyses.*

**Author's response:** dB is a unit to measure the relative magnitude of energy. The method is proposed by ALEXANDER B. RABINOVICH, and detailed description of the method can be found in (Rabinovich, 2009). We use different units on the left and right ordinates of each sub-figure, i.e. second and minute respectively, to presnt the results in different unit ordinates. We conduct wavelet analysis for longer original waveform and wider period band than presented, to avoid blanked-out peripheral area.

*28.How Eq. 4 is obtained?*

**Author's response:** Thanks, we've added a citation (Rabinovich, 1997) to explain the equation, through which you can find the detailed derivation of this equation.

*29.Lines 361-362, How are these factors specified from the present study?*

**Author's response:** The influence conditions are mentioned in the second paragraph in section 4.1. We paste the related part here: "The long-traveling capability could be associated with the ~ 10000 m deep water depth of the Tonga Trench that keeps the source signals from substantial attenuation. In deep open ocean, the wavelength of a tsunami can reach two hundred kilometers, but the height of the tsunami may be only a few centimeters. Tsunami waves in the deep ocean can travel thousands of kilometers at high speeds, meanwhile losing very little energy in the process. The long oscillation can be attributed to the multiple reflections of the incoming waves trapped in the shallow-water bay at Charleston."

*30.10b, atmospheric and tsunami wave forms are also not mentioned in the context, these sub-figures could be deleted.*

**Author's response:** Fig. 10b is mentioned in the second paragraph in section 4.2.

*31.Lines 374-378, Lamb wave speed is rather fast, even it circles the earth multiple times, it should not or less contribute to the 3 days tsunami event, especially considering that after circling, the Lamb wave energy decays.*

**Author's response:** Yes, we agree with your opinion which is also suggested by our results.

*32.Line 379, what is the meaning of resonance between ocean and atmospheric waves? They have very much different frequency, how can resonance between these two be triggered?*

**Author's response:** Some atmospheric gravity wave modes have velocities close to that of tsunami waves in most parts of the Pacific Ocean, which results in resonance effect (Kubota et al., 2022).

*33.Line 380, What is the difference among atmospheric gravity wave, Lamb wave, and shock wave? These concepts must be clarified in the context.*

**Author's response:** Thanks, we have added explanations in the Introduction section to clarify the concepts.

**Author's change to manuscript:** The added sentences in the Introduction section: "Atmospheric waves propagating in the atmospheric fluid with different speeds are generated by different physical mechanisms (E. E. Gossard & W. H. Hooke, 1975). Nonlinearities in the process may lead to the formation of shock-wave and period lengthening. The balance between gravity and buoyancy causes gravity waves. The acoustic wave propagate by atmospheric fluid compression and rarefaction (Matoza et al., 2022)."

*34.Line 385, Please specify what kind of tsunami speed here mentioned?*

**Author's response:** To make the logic of this part smoother, we've modified the context. We first explain the definition of Proudman effect and then mention the tsunami speeds more specifically.

**Author's change to manuscript:** We've modified the related text as: "Second, when Lamb wave speed approaches the tsunami speed, Proudman resonance gradually increase tsunami heights, wherein Proudman resonance optimally maximizes tsunami heights when they match well. In deep oceanic trenches, such as Mariana and Tonga-Kermadec trench (10000–11000 m), tsunami velocities range between ~314–330m/s which are comparable with those of the observed Lamb waves 315–340 m/s."

*35.Line 387, Proudman resonance is a well-known and old concept. No need to refer 2022 papers. Why continuously? the deep trench is generally rather narrow, while the Lamb wave speed is very fast and it only need short duration for Lamb wave passing through the trench.*

**Author's response:** Thanks, we've removed the word 'continuously' as suggested and put the citations of the 2022 papers in a more appropriate place. The explanation of possible contribution of the Proudman effect is largely based on its theoretical definition. Since in some of the deep oceanic trenches, tsunami velocity could range between ~314–330m/s which are comparable with that of the observed Lamb waves 315–340 m/s, we can't exclude its contribution to the tsunami oscillation, theoretically.

**Author's change to manuscript:** We've rephrased the sentence as: "Therefore, the resonance effect could be a possible source of increased wave energy, especially in the deep trenches (Lynett et al., 2022; Tanioka et al., 2022)."

*36.Lines 393-394, Why a 2005 paper is referred for the 2021 Tonga event??*

**Author's response:** Thanks for pointing out this mistake. We have replaced the reference with a corrected one.

**Author's change to manuscript:** "…fast-moving atmospheric waves for the Mw 5.8 volcanic eruption (Matoza et al., 2022) …"

*37.Line 396, only 70 km from the source??*

**Author's response:** The maximum runup of the 2011 Tohoku earthquake is measured at Miyako in the Iwate Prefecture, a coastal port ~70 km away from the epicenter.

*38.Line 429-430, Resonance effect can only amplify the tsunami height, no the duration. The description here is not serious.*

**Author's response:** Thanks for pointing out the unserious description. We've changed our description here.

**Author's change to manuscript:** " we do not have clear evidence that atmospheric acoustic-gravity waves from the 2022 HTHH eruption directly contribute to the long-lasting tsunami, but the resonance effect associated with ocean waves could a possible source of increased wave energy."

*39.Lines 433-435, Fig. 11 indicate that the long-lasting of HTHH tsunami is not related to the Lamb wave induced tsunamis, but related to the subsequently gravity wave and its interaction with the coastal bathymetry and coastal configuration. In other words, interactions between the Lamb wave induced tsunamis and coastal bathymetry/coastal configuration are negligible.*

**Author's response:** Thanks. Based on our analysis, we think the long tsunami duration is indirectly from the contribution of air-sea coupling with atmospheric acoustic-gravity waves (including shock wave, Lamb wave, gravity wave), but the interaction of local bathymetric characteristics with the ocean waves left by each passage of atmospheric acoustic-gravity waves. The comparison of hydrodynamic characteristics between the 2022 HTHH tsunami event and the 2011 Tohoku tsunami event suggests the volcanic tsunami oscillation was prolonged by their interplays with local bathymetry.

*40.Please specify the meanings of different white dashed lines in Fig. 11 caption.*

**Author's response:** Thanks, done as suggested.

**Author's change to manuscript:** We've added a sentence as: "Horizontal white dashed lines respectively mark reference periods of 10 min and 30 min. "

*41.Lines 454-455, Generally, I do not think Proudman resonance from the Lamb wave is the reason for the large coastal tsunami height since the ocean is still too shallow and the deep trench only exists within a narrow area being generally perpendicular to the tsunami propagation direction.*

**Author's response:** Thanks. We agree that Proudman resonance from the Lamb wave is not the reason for the large coastal tsunami height. Regarding the contribution of Proudman resonance to the tsunami event, please kindly refer to our response to comment 35.

*42.Lines 456-463, There have been well-known from the previous studies. The trapping effect in the coastal region should be considered for tsunami warning, e.g., edge wave and so on. The resonance effect can only amplify the tsunami wave height, which may indirectly leads to the long lasting of tsunami event.*

**Author's response:** Thanks. When the oscillation periods of ocean wave and local bathymetry (bays/ harbors or the continental shelves) match, the resonance effect between ocean waves and local bathymetry form. The effect can not only amplify the tsunami height, but also prolong the tsunami duration (Satake et al., 2020;Wang et al., 2021). The long duration is produced by the reflection and interference of tsunami waves at the edge of bays/

harbors or the continental shelves. For example, following the main tsunami arrival, a series of waves reflect from the shelf edge back to the coast repeatedly. These repeated reflections not only trap the tsunami energy from entering the deep ocean, but also constitute shelf resonance (Rabinovich, 2009). Similarly, in harbor/bay case, incident waves reflect back from the end of the bay and reach the entrance repeatedly(Miles, 1974). Such reflections prolong the duration of tsunami events.

*43.Conclusions should be amended following the aforementioned comments.*

**Author's response:** Thanks, we have revised the manuscript accordingly.

**Author's change to manuscript:** We've modified the related context in the conclusion: "Although the resonance effect with the atmospheric acoustic-gravity waves theoretically could be a source of increased wave energy, its direct contribution to the long-lasting oscillation is not demonstrated yet. However, the comparison of hydrodynamical characteristics between the 2022 HTHH tsunami event and the 2011 Tohoku tsunami event well demonstrated that the interactions between the ocean waves left by atmospheric waves and local bathymetry contribute to the long-lasting Pacific oscillation of the 2022 tsunami event."

**Reference**

Amores, A., Monserrat, S., Marcos, M., Argüeso, D., Villalonga, J., Jordà, G., & Gomis, D. (2022). Numerical simulation of atmospheric Lamb waves generated by the 2022 Hunga-Tonga volcanic eruption. *Geophysical Research Letters*, *49*, e2022GL098240. https://doi.org/10.1029/2022GL098240

Cohen, M. X. (2018). A better way to define and describe Morlet wavelets for time-frequency analysis. https://doi.org/10.1101/397182

Doğan, G. G., Yalçıner, A. C., Yuksel, Y., Ulutaş, E., Polat, O., Güler, I., et al. (2021). The 30 October 2020 Aegean Sea Tsunami: Post-Event Field Survey Along Turkish Coast. *Pure and Applied Geophysics*, *178*, 785–812. https://doi.org/10.1007/s00024-021-02693-3

E. E. Gossard, & W. H. Hooke. (1975). Waves in the Atmosphere: Atmospheric Infrasound and Gravity Waves—Their Generation and Propagation. *Elsevier*.

Heidarzadeh, M., & Satake, K. (2013). Waveform and Spectral Analyses of the 2011 Japan Tsunami Records on Tide Gauge and DART Stations Across the Pacific Ocean. *Pure and Applied Geophysics*, *170*, 1275–1293. https://doi.org/10.1007/s00024-012-0558-5

Kubota, T., Saito, T., & Nishida, K. (2022). Global fast-traveling tsunamis by atmospheric pressure waves on the 2022 Tonga eruption. *Science*. https://doi.org/10.1126/science.abo4364

Lynett, P., McCann, M., Zhou, Z., Renteria, W., Borrero, J., Greer, D., et al. (2022). The

Tsunamis Generated by the Hunga Tonga- Hunga Ha ' apai Volcano on January 15 , 2022. *ResearchSquare*. https://doi.org/10.21203/rs.3.rs-1377508/v1

Matoza, R. S., Matoza, R. S., Fee, D., Assink, J. D., Iezzi, A. M., Green, D. N., et al. (2022). Atmospheric waves and global seismoacoustic observations of the January 2022 Hunga eruption ,Tonga. *Science*. https://doi.org/10.1126/science.abo7063

Miles, J. (1974). Harbor seiching. *Annual Review of Fluid Mechanics*, (1686), 17–35.

Nishikawa, Y., Yamamoto, M., Nakajima, K., Hamama, I., Saito, H., & Kakinami, Y. (2022). What excited tsunami from Tonga 2022 eruption ? Observation and theory. *ResearchSquare*, (April). https://doi.org/10.21203/rs.3.rs-1513574/v1

Parker, B. B. (2007). Tidal analysis and prediction. *Silver Spring, MD, NOAA NOS Center for Operational Oceanographic Products and Services, 378pp (NOAA Special Publication NOS CO-OPS 3)*. https://doi.org/10.25607/OBP-191

Rabinovich, A. B. (1997). Spectral analysis of tsunami waves: Separation of source and topography effects. *Journal of Geophysical Research: Oceans*, *102*(C6), 12663–12676. https://doi.org/10.1029/97JC00479

Rabinovich, A. B. (2009). Seiches and harbor oscillations. in: Handbook of coastal and ocean engineering, *pp*, 193–236.

Satake, K., Heidarzadeh, M., Quiroz, M., & Cienfuegos, R. (2020). History and features of trans-oceanic tsunamis and implications for paleo-tsunami studies. *Earth-Science Reviews*, *202*, 103112. https://doi.org/10.1016/j.earscirev.2020.103112

Tanioka, Y., Yamanaka, Y., & Nakagaki, T. (2022). Characteristics of the deep sea tsunami excited offshore Japan due to the air wave from the 2022 Tonga eruption. *Earth, Planets and Space*, *74*, 61. https://doi.org/10.1186/s40623-022-01614-5

Wang, Y., Zamora, N., Quiroz, M., Satake, K., & Cienfuegos, R. (2021). Tsunami Resonance Characterization in Japan Due to Trans-Pacific Sources: Response on the Bay and Continental Shelf. *Journal of Geophysical Research: Oceans*, *126*(6), 1–16. https://doi.org/10.1029/2020JC017037

---

## Author Response (AR2)

October 10th, 2022

Dear Dr. González,

We sincerely thank you and the two reviewers for the encouragements and constructive comments you have made on the manuscript. We have addressed the questions and highlighted the areas where changes are made. Here we present our point-by-point responses and revisions to the comments.

Linlin Li

Note: The comments are in "*italics*", and our responses and revisions are in "regular" text (in blue) for clarity.

═══════════════════════════════════════════════════════════════

**Response to Reviewer #1**

**General Comments:**
* * *
**Author's response:** We thank referee #1 for the encouragements he/she made on the manuscript.

═══════════════════════════════════════════════════════════════

**Response to Reviewer #2**

*1.Response to comment 1, 'the arrival of Lamb wave from the waveforms recorded by DARTs and barometers'. Wrong concept. Here, Lamb wave only stands for the atmospheric sound wave, which could only be detected/measured by barometers. What DARTs detected/measured is the Lamb wave-induced water surface wave/tsunamis.*

**Author's response:** Thanks for pointing out the unserious description. We have made corresponding changes to the related contents in the manuscript.

**Author's change to manuscript:** We have changed the related parts as "arrival times of Lamb wave-induced tsunami" or "arrivals of Lamb wave-induced tsunami".

*2. Response to comment 2, How about cut-off frequency of 6h or 10 h? Applying different cut-off frequencies could lead to significant differences or not? BTW, the two examples in the response are for the earthquake induced tsunamis, different from the present Tonga one.*

**Author's response:** Thanks. Using different cut-off frequencies such as 10 h, 6h or 3h doesn't change the results because the main period of the tsunami waves range concentrate in period band of ~2-100 min. Since the period bands are in the similar range with those of the earthquake-induced tsunami waves, the de-tided method should be suitable for this tsunami

event.

*3. Response to comment 3, Why 0.2 m? not 0.3 m or 0.1 m? BTW, many recorded tsunami heights in the coastal area are just around 0.2 m during the Tonga tsunami event, similar to this cut-off height.*

**Author's response:** Thanks. Indeed, there is a trade-off between keeping more gauges with smaller cut-off height and maintain the dataset with relatively good quality. Choosing 0.2 m as the cut-off height will miss some tide gauges with wave height less than 0.2m, but the chosen gauges are reasonably distributed in the Pacific Ocean which we believe could well demonstrate our key points. The choice of 0.1m will keep more unusable tide gauges with noise height larger than tsunami signal, so that we had to manually remove massive data.

*4. Response to comment 4, Definition of 'wave amplitude' is the half of wave height. The ordinate should be 'Normalized water level', not 'Normalized wave amplitude'. Please also specify how the normalization (with respect to the local mean water level or others?) was conducted in the context or caption.*

**Author's response:** Thanks for pointing out this mistake. We have changed the ordinate as suggested. The data are normalized with respect with the largest amplitude of each tide gauge.

**Author's change to manuscript:** We have changed the ordinate label as 'Normalized water level' and added a sentence in the caption as "The data are normalized with respect to the largest amplitude of each tide gauge.".

[Figure]

**Figure 2. Detided tsunami waveforms at (a) DART buoys and (b) tide gauges. Waveforms in both subplots are shown in ascending distance. Azi stands for azimuth. The data are normalized with respect to the largest amplitude of each tide gauge.**

*5. Response to comment 6, Mother function selection is important for wavelet analysis leading to different results. Although Morlet is frequently used in wavelet analysis, this mother function is not universal. Selection of the mother function should be case-dependent with respect to the temporal characteristics of the original signal.*

**Author's response:** Thanks. The mother wavelet governs how the wavelet transform transfers time information into the frequency domain. The mother wavelet we use is the Morlet wavelet, which is optimally suitable for identifying the oscillatory components of wave signals (PS., 2002). Previous applications of the Morlet function demonstrate that it works well in tsunami period analysis. The method has been widely applied in many tsunami

cases for detecting the tsunami period information, such as tsunamis generated by two successive earthquake (Mw 7.4 and Mw 8.1 earthquakes) in the Kermadec Islands on 4 March 2021 (Wang et al., 2022a), and tsunami of 2022 Tonga volcanic eruption in in Lingding Bay, China (Wang et al., 2022b).

*6. Response to comment 7, As mentioned by the authors, Lamb wave has a period of 30-40 min, thus the induced tsunami waves. Using a 20 min moving time window can exclude the 30 min Lamb wave induced tsunami component?*

**Author's response:** We apologize for this misunderstanding caused by our previous expression. In our previous replies, we used the word "exclude" in this sentence*: "To achieve this purpose, we need to choose a representative wave period that covers most tsunami waves with significant amplitude and long-lasting time duration, as the moving time window. With the criterion, we exclude the periods of tsunami components from shock wave, Lamb wave and conventional seafloor crustal displacement, because they are either distributed in a limited time period in each waveform, or have very limited spatial and temporal impact, therefore, do not meet the requirements."* We actually meant to say that we choose not to use the periods of… Using a 20 min moving time window can not exclude the 30 min Lamb wave induced tsunami component. Sorry for this misleading expression.

*7. Response to comment 11, I am thinking that the measurement error of DART buoy should be even larger than 0.1 mm.*

**Author's response:** Thanks. We have tested some other criterion values in the range of one order of magnitude difference (0.1mm, 0.5mm and 1mm ) to capture the arrival time. We notice larger criterion leads a visible shift of the arrival time from the time where wave amplitude starts to increase. Since, the recorded maximum amplitude of the most DARTs is only a few centimeters in the event, we decided to choose the relatively smaller value. Additionally, according to a description of DART II given by PMEL, the Dart buoy is quite sensitive to tiny change of the sea bottom pressure ($\sim 2*10^{-7}$ m, (Meinig et al., 2005).

Table 1: DART II performance characteristics

| Characteristic | Specification |
| --- | --- |
| Reliability and data return ratio: | Greater than 80% |
| Maximum deployment depth: | 6000 meters |
| Minimum deployment duration: | Greater than 1 year |
| Operating Conditions | Beaufort 9 (survive Beaufort 11) |
| Maintenance interval, buoy | Greater than 2 years |
| Maintenance interval, tsunameter | Greater than 4 years |
| Sampling interval, internal record: | 15 seconds |
| Sampling interval, event reports: | 15 and 60 seconds |
| Sampling interval, tidal reports: | 15 minutes |
| Measurement sensitivity: | Less than 1 millimeter in 6000 meters; $2 \times 10^{-7}$ |
| Tsunami data report trigger | Automatically by tsunami detection algorithm |
|  | On-demand, by warning center request |
| Reporting delay: | Less than 3 minutes |
| Maximum status report interval: | Less than 6 hours |

*8. Response to comment 12, In this study, authors assumed the temperature is for low*

*elevation. While, I am not sure temperature in Eq. (2) originally proposed by Gossard and Hooke (1975) is for low or high elevation since Lamb wave is an atmospheric wave which should be elevation related.*

**Author's response:** Thanks. The equation is a solution of the momentum equations with zero vertical velocity, meaning that Lamb waves have purely horizontal motion, occupying the full depth of the troposphere and with a maximum pressure signal at the surface. These waves are only slightly affected by the Earth's rotation and travel at the speed of sound in the media (Gossard and Hooke, 1975). Therefore, the equation is not elevation related. The equation has been successfully applied in numerical simulation of air-sea coupling with Lamb waves of 2022 Tonga eruption (Amores et al., 2022).

*9. Response to comment 14, These black lines are very much sensitive. As shown in Fig. 3, these lines, in fact, do not fit with the arrival times of the Lamb waves (black dots).*

**Author's response:** The black lines are only used as a visual reference to help reader understand the velocity, not to fit the arriving time points. For example, in the third subplot, we can see most balck dotes are located between the lines of 320 m/s and 310 m/s meanwhile around the line of 315 m/s.

*10. Response to comment 15, No bathymetry information in Chanaral Bay?*

**Author's response:** Thanks. We have changed the color bar of bathymetry in Chanaral Bay in Figure S1 to present the bathymetric data more clearly.

[Figure]

**Figure S1. Bathymetry in Japan (a) and Chañaral bay (b) in Chile. Bathymetry data is downloaded from GEBCO (http://www.gebco.net ).**

*11. Response to comment 17, In this sense, the ~10 h delays in New Zealand, Hawaii, and west coast of America are from which mechanism? edge wave? resonance? reflection?*

**Author's response:** Thanks. It's hard to pinpoint which mechanism play a dominant role in a specific location. It requires more detailed analysis. In order to further investigate which mechanism is responsible for a specific location, well targeted modelling and field survey are needed. For example, through careful tsunami modelling and waveform analysis, (Rabinovich

et al., 2006) suggest that most tsunami energy of the 2005 California tsunami and 1994 Shikotan tsunami in the western coast of America is trapped in the edge waves and propagated along the long coast with little energy loss. The trapped energy and the waveguide are mainly influenced by the irregular coastal shelves of the western coast of America.

*12. Response to comment 18, Previous observations in Heidarzadeh & Satake (2013) should be generally from earthquake induced tsunami, the first wave thus is the maximum one in deep ocean (DART). However, Tonga tsunami was not induced by earthquake. Accordingly, the first wave was not the maximum one. Please do not mix various concepts.*

**Author's response:** Thanks for the reminder, we have modified the related text to present the result in a more logical way.

**Author's change to manuscript:** We have modified the related content in the manuscript as "On the other hand, for tsunami events with earthquake origins (e.g. Heidarzadeh and Satake, 2013), the first waves recorded by DART buoys are normally observed as the largest wave since DART buoys are located in the deep sea and less influenced by bathymetric variation. In the case of Tonga tsunami event, we observe …."

*13. Response to comment 25, If Fig. 7 could not support the discussion, it could be deleted. Anyway, discussions about the shock wave are not persuaded.*

**Author's response:** Thanks for the suggestion. We actually have thought seriously about whether we should keep this part. We finally decided to keep the shock wave related discussion as it might have certain value for tsunami hazard research if such mechanism indeed exist. Although it is not that significant compared with the conspicuous Lamb wave-induced tsunami, the wave in New Zealand (Two columns on the left in Figure 7) is more conspicuous than those in the rest of the Pacific station (Two columns on the right in Figure 7). We think such observation is worth being mentioned and hopefully it can attract more attention to research the complex air-sea coupling mechanisms of fast-arriving tsunami waves.

*14. Response to comment 26, My comment is that descriptions in Lines 285-286 are not suitable since energy distribution in Figs. 5, 6, and 8 are in fact not consistent with each other.*

**Author's response:** Thanks. I agree with you that the three figures are not consistent in terms of the energy distribution. But here we only refer to the wave periods shown between the solid red line and dashed white line in Figure 6 and Figure 8. The consistency is reflected by both arrival times and wave periods.

``*15. Response to comment 27, Could not find the definition of 'dB' in the mentioned paper. Please specify clearly in the response. No need to use different units for left and right ordinates, which contaminates the proper understanding of the figure. Could authors show the original results with the blanked-out peripheral area of the spectrum? Why need to avoid the blanked-out peripheral area?*

**Author's response:** Thanks. Decibel (dB) is calculated from: $dB = 10 \log(A/A_0)$, where A is wavelet power, $A_0$ is a reference wavelet power (Thomson and Emery, 2014). In this case, I choose maximum wavelet power as reference $A_0$. According to your suggestion, we have

reconducted and updated all the results of wavelet analysis to single ordinate, including Figures 5, 6, 8, 9 and 11. We also have the figures updated to the original results with the blanked-out peripheral areas of the spectrum. Please refer to these figures below

**Author's change to manuscript**:We have added the definition of dB in related content as: "Decibel (dB) is calculated from: dB = 10 log(A/A0), where A is wavelet power, A0 is a reference wavelet power of the maximum one (Thomson and Emery, 2014)."

| **Previous** Figures 5, 6, 8, 9 and 11 | **Updated** Figures 5, 6, 8, 9 and 11 |
|---|---|

[Figure]

[Figure]

*16. Response to comment 28, Authors should answer this question directly, rather than leaving it to Rabinovich (1997).*

**Author's response:** Thanks. This is a simplified model to build a quantitative relationship between source typical dimension (L) and tsunami period (T). The average depth (H) of the source area is used to calculate initial propagation velocity through $\sqrt{gH}$ (g is gravitational

acceleration). The observed periods times the velocity equals initial periodic wavelength. The tsunami source only displaces half wavelength of water and another half is a result of water volume conservation. Therefore, the typical source dimension is only half of the initial periodic wavelength, forming the equation $L = \frac{T\sqrt{gH}}{2}$.

*17. Response to comment 32, This is definitely wrong. To trigger the Proudman resonance, a very larger water depth is needed to make the tsunami propagation speed being close to the Lamb wave speed. This needs a water depth of about 10000 m, which is not prevail in the Pacific Ocean with an averaged water depth of 4000 m.*

**Author's response:** Thanks. Yes, we agree with you that the Proudman resonance plays a negligible role in this event. We have adjusted our previous writing to clarify our point. The revised manuscript mainly emphasizes the possible contribution of the atmospheric gravity waves with slower propagation speed (~200-245 m/s). Besides the Lamb wave, Watanabe et al., 2022 detected internal Pekeris wave which propagate with a slower horizonal phase speed of ~245 m/s and gravity waves with even slower propagation speed by analyzing radiance observations taken from the Himawari-8 geostationary satellite. Atmospheric waves with such speeds are more likely to resonant with the conventional tsunami waves and provide continuous energy supply (Kubota et al., 2022)..

**Author's change to manuscript:** We've modified the related text as: "The resonance effects between ocean waves and atmospheric waves could contribute to the long oscillation on coastlines. Besides the Lamb wave, Watanabe et al., 2022 detected internal Pekeris wave which propagate with a slower horizonal phase speed of ~245 m/s and gravity waves with even slower propagation speed by analyzing radiance observations taken from the Himawari-8 geostationary satellite. Atmospheric waves with such speeds are more likely to resonant with the conventional tsunami waves and provide continuous energy supply (Kubota et al., 2022).

*18. Response to comment 33, My question is to clarify the basic concepts of gravity wave, Lamb wave, and shock wave in the MS, rather than 'acoustic wave' and so on. In addition, should be 'Gossard and Hooke, 1975', not 'E.E. Gossard and W.H. Hooke, 1975'.*

**Author's response:** Thanks. We have modified the part of content and present basic concepts of gravity wave, Lamb wave, and shock wave in the MS.

**Author's change to manuscript:** The added sentences in the Introduction section: "Atmospheric waves propagating in the atmospheric fluid are generated by different physical mechanisms (Gossard and Hooke, 1975). Lamb wave is a horizontally propagating acoustic waves in Lamb mode which is trapped at the earth's surface with group velocities close to the mean sound velocity of the lower atmosphere (e.g. Lamb, 1932). Atmospheric gravity wave is triggered when air molecules in the atmosphere are disturbed vertically other than horizontally (e.g. Le Pichon et al., 2010). Nonlinear propagation of atmospheric wave may cause period lengthening and the formation of shock-wave (Matoza et al., 2022). "

*19. Response to comment 34, Since the trench is very much narrow and the tsunami propagation speed is very fast, there is no time for Proudman resonance to be fully functional to significantly increase the tsunami wave height.*

**Author's response:** Thanks. We have modified this paragraph and the updated content added the contribution from the Pekeris resonant resonance mode. Please also refer to our response to comment 17.

**Author's change to manuscript:** We've modified the related text as: "The resonance effects between ocean waves and atmospheric waves could contribute to the long oscillation on coastlines. Besides the Lamb wave, Watanabe et al., 2022 detected internal Pekeris wave which propagate with a slower horizonal phase speed of ∼245 m/s and gravity waves with even slower propagation speed by analyzing radiance observations taken from the Himawari-8 geostationary satellite. Atmospheric waves with such speeds are more likely to resonant with the conventional tsunami waves and provide continuous energy supply (Kubota et al., 2022). "

*20. Response to comment 35, Appreciate if authors could present some new stuffs, rather than repeating some old understandings.*

**Author's response:** Thanks. Please kindly refer to our response to comment 17 and 19.

*21. Response to comment 37, Must be larger than 70 km. Iwate Prefecture is not so near to the epicenter. Please double check this.*

**Author's response:** Thanks for the reminder. The 70 km we mentioned is the distance between epicenter and the eastern coast of Japan's Tohoku region. The largest run-up is observed in Tohoku's Iwate Prefecture, specifically at Miyako. To avoid confusion, we have modified the related content in the manuscript to make.

**Author's change to manuscript:** We have corrected the sentence as "In the near-field, the 2011 Tohoku earthquake produced runup up to 40 m at Miyako in the Iwate Prefecture in Japan's Tohoku region (Mori et al., 2011). The epicenter is approximately 70 km east coast of the Oshika Peninsula of Tohoku region."

*22. Response to comment 39, What I want to point out here is that the long-lasting tsunami of 2022 HTHH and 2011 Tohoku event should come from the similar mechanisms, i.e., interactions between gravity wave and coastal bathymetry. Interactions between Lamb wave induced tsunamis and coastal bathymetry could be neglected.*

**Author's response:** Yes, we agree with you.

*23. Response to comment 42, This sentence is strange. What is the meaning of 'oscillation period of local bathymetry'? How can bathymetry oscillate? Please be serious with context description. In addition, do not talk about too many old stuffs, such as edge wave, reflection and so on. Appreciate some new understandings from the Tonga event.*

**Author's response:** Thanks. "Oscillation period of local bathymetry" should be "free oscillation period of the local bathymetry". The free oscillation in a near-shore region possesses some eigen-modes of natural oscillations. The modes are closely associated with coastal geometry and bathymetry. We have checked the associated context in the manuscript and modified the description.

It's true that the potential hazard caused by edge waves, reflection and … is relatively well known. But the interesting observation we have here is the local effect still play a dominant role in tsunami behaviors even the driven forces are distinctly different (volcanic tsunami and

seismogenic tsunami). We think it's the first time we can demonstrate such similarity with such abundant instrumental records of such a rare global volcanic tsunami event.

**Reference**

Amores, A., Monserrat, S., Marcos, M., Argüeso, D., Villalonga, J., Jordà, G., and Gomis, D.: Numerical simulation of atmospheric Lamb waves generated by the 2022 Hunga-Tonga volcanic eruption, Geophys. Res. Lett., 49, e2022GL098240, https://doi.org/10.1029/2022GL098240, 2022.

Gossard, E. E. and Hooke, W. H.: Waves in the Atmosphere: Atmospheric Infrasound and Gravity Waves—Their Generation and Propagation, Elsevier, 1975.

Heidarzadeh, M. and Satake, K.: Waveform and Spectral Analyses of the 2011 Japan Tsunami Records on Tide Gauge and DART Stations Across the Pacific Ocean, Pure Appl. Geophys., 170, 1275–1293, https://doi.org/10.1007/s00024-012-0558-5, 2013.

Kubota, T., Saito, T., and Nishida, K.: Global fast-traveling tsunamis by atmospheric pressure waves on the 2022 Tonga eruption, Science (80-. )., https://doi.org/10.1126/science.abo4364, 2022.

Lamb, H.: Hydrodynamics, Cambridge Univ. Press, 1932.

Matoza, R. S., Matoza, R. S., Fee, D., Assink, J. D., Iezzi, A. M., Green, D. N., Kim, K., Lecocq, T., Krishnamoorthy, S., Lalande, J., Nishida, K., and Gee, K. L.: Atmospheric waves and global seismoacoustic observations of the January 2022 Hunga eruption ,Tonga, Science (80-. )., https://doi.org/10.1126/science.abo7063, 2022.

Meinig, C., Stalin, S. E., and Nakamura, A. I.: Real-Time Deep-Ocean Tsunami Measuring, Monitoring, and Reporting System: The NOAA DART II Description and Disclosure, NOAA Pacific Mar., 2005.

Mori, N., Takahashi, T., Yasuda, T., and Yanagisawa, H.: Survey of 2011 Tohoku earthquake tsunami inundation and run-up, Geophys. Res. Lett., 38, L00G14, https://doi.org/10.1029/2011GL049210, 2011.

Le Pichon, A., Blanc, E., and Hauchecorne, A.: Infrasound monitoring for atmospheric studies, Springer Science & Business Media, 1–735 pp., https://doi.org/10.1007/978-1-4020-9508-5, 2010.

PS., A.: The Illustrated Wavelet Transform Handbook, Philadelphia, PA Inst. Phys. Publ., 2002.

Rabinovich, A. B., Stephenson, F. E., and Thomson, R. E.: The California tsunami of 15 June 2005 along the coast of North America, Atmos. - Ocean, 44, 415–427, https://doi.org/10.3137/ao.440406, 2006.

Thomson, R. E. and Emery, W. J.: Data Analysis Methods in Physical Oceanography: Third Edition, New York: Elsevier, 1–716 pp., 2014.

Wang, Y., Heidarzadeh, M., Satake, K., and Hu, G.: Characteristics of two tsunamis generated by successive Mw 7.4 and Mw 8.1 earthquakes in Kermadec Islands on March 4,2021, Nat. Hazards Earth Syst. Sci., 22, 1–10, https://doi.org/10.5194/nhess-2021-369,

2022a.

Wang, Y., Wang, P., Kong, H., and Wong, C.-S.: Tsunamis in Lingding Bay, China, caused by the 2022 Tonga volcanic eruption, Geophys. J. Int., ggac291, https://doi.org/10.1093/gji/ggac291, 2022b.

Watanabe, S., Hamilton, K., Sakazaki, T., and Nakano, M.: First Detection of the Pekeris Internal Global Atmospheric Resonance: Evidence from the 2022 Tonga Eruption and from Global Reanalysis Data, J. Atmos. Sci., 79, 3027–3043, https://doi.org/10.1175/jas-d-22-0078.1, 2022.

---

## Author Response (AR3)

January 9th, 2023

Dear Dr. González,

We sincerely thank you and referee #2 for the constructive comments. We have addressed the questions and highlighted the areas where changes are made. Here we present our point-by-point responses and revisions to the comments.

Linlin Li

Note: The comments are in "*italics*", and our responses and revisions are in "regular" text (in blue) for clarity.

═══════════════════════════════════════════════════════════════

**Response to Reviewer #2**

*1. Response to comment 7 (original response to comment 11). Specification in this table indicates the measurement sensitivity is 1 mm. Hence, 0.1 mm could be the measurement error of DART. Even in lab experiment, it is hard to achieve an accuracy of 0.1 mm, not to mention the field measurement of DART.*

**Author's response:** Thank you. It's a bit subjective when selecting a criterion value to identify the tsunami arrival time. In addition to the criterion of 0.1 mm. We have tested other criterion values such as 1 mm and made a careful comparison of different results. When we zoom in the figures, we find the larger criterion leads a visible shift of the arrival time at some Dart stations. However, the quantitative arrival difference between the results of 1 mm and 0.1 mm range between only a few seconds and ~4 min. To present the visual shift of the arrival time, we zoom in on the subplot (c) of both Criterions, for example, arrival time in the blue circle shift to the right ~3 min. The different results of the two criterions are shown in the figures below. We therefore conclude that the selection of criterion values (0.1mm or 1mm) does not affect the interpretation of the results.

[Figure]

[Figure]

Zoom in on the subplot (c) of Criterion = 0.1 mm

Zoom in on the subplot (c) of Criterion = 1 mm

[Figure]

*2. Response to comment 9 (original response to comment 14). In last response, authors mentioned that 'We set the velocity to fit the Lamb wave arrival', while in this response, authors mentioned that 'The black lines are only used as a visual reference to help reader understand the velocity, not to fit the arriving time points'. I am lost with these two responses. Since the difference among the three velocities in sub-figures is not that large, i.e., 340, 325, and 315 m/s, plotting these black lines should be very much sensitive. Otherwise, the conclusions may change.*

**Author's response:** We use the black lines which represent different constant velocities, to identify the velocity range of the Lamb wave-induced tsunami waves. The previous word 'fit' might be a bit confusing. We have rephrased this part to clarify our approach in the related content.

**Author's change to manuscript:** We have modified the related content in the manuscript as "Using different velocity values as constraints, we illustrate …"  "Figure 3. Identifying the Lamb wave-induced tsunami velocities using different constant velocities as constraints"

*3. Response to comment 15 (original response to comment 27). The blanked-out peripheral area of the spectrum shown in the response is different from what I know. Authors may refer to: Grinsted, A., Moore, J.C. and Jevrejeva, S. (2004) Application of the cross wavelet transform and wavelet coherence to geophysical time series. Nonlinear Process. Geophys., 11, 561–566.*

**Author's response:** Thank you. This study adopts one of the wavelet methods, which is called "cross wavelet analysis and wavelet coherence", to analyze the waveforms of two time series together (Grinsted1 et al., 2004). The blanked-out peripheral area you mentioned is 5% significance level against noise backgrounds and the cone of influence where edge effects might distort the picture. It's one of the features of this method. We believe the blanked-out peripheral area doesn't influence how we interpret the results. Similar approach has been used by many previous studies (e.g. Titov et al., 2005; Wang et al., 2022; Heidarzadeh and Satake, 2013)

**Reference**

Grinsted1, A., Moore1, J. C., and Jevrejeva, S.: Application of the cross wavelet transform and wavelet coherence to geophysical time series, Nonlinear Process. Geophys., 11, 515–533, https://doi.org/10.5194/npg-11-515-2004, 2004.

Heidarzadeh, M. and Satake, K.: Waveform and Spectral Analyses of the 2011 Japan Tsunami Records on Tide Gauge and DART Stations Across the Pacific Ocean, Pure Appl. Geophys., 170, 1275–1293, https://doi.org/10.1007/s00024-012-0558-5, 2013.

Titov, V., Rabinovich, A. B., Mofjeld, H. O., Thomson, R. E., and Gonza, F. I.: The Global Reach of the 26 December 2004 Sumatra Tsunami, Science (80-. )., 309, 2045–2049, https://doi.org/10.1126/science.1114576, 2005.

Wang, Y., Heidarzadeh, M., Satake, K., and Hu, G.: Characteristics of two tsunamis generated by successive Mw 7.4 and Mw 8.1 earthquakes in Kermadec Islands on March 4,2021, Nat. Hazards Earth Syst. Sci., 22, 1–10, https://doi.org/10.5194/nhess-2021-369, 2022.